# Physics-Informed Inductive Biases for Voltage Prediction in Distribution Grids

## Abstract

Voltage prediction in distribution grids is a critical yet difficult task for maintaining power system stability. Machine learning approaches, particularly Graph Neural Networks (GNNs), offer significant speedups but suffer from poor generalization when trained on limited or incomplete data. In this work, we systematically investigate the role of inductive biases in improving a model's ability to reliably learn power flow. Specifically, we evaluate three physics-informed strategies: (i) power-flow-constrained loss functions, (ii) complex-valued neural networks, and (iii) residual-based task reformulation. Using the ENGAGE dataset, which spans multiple low- and medium-voltage grid configurations, we conduct controlled experiments to isolate the effect of each inductive bias and assess both standard predictive performance and out-of-distribution generalization. Our study provides practical insights into which model assumptions most effectively guide learning for reliable and efficient voltage prediction in modern distribution networks.

## 1 Introduction

Distribution networks are shifting from passive delivery systems to actively managed infrastructures shaped by high DER penetration, electrification, and frequent reconfigurations. These conditions create volatile conditions and tight real-time decision windows. Voltage prediction is a central task in this setting because maintaining voltages within operational limits is essential for efficiency, equipment safety, and system stability. For an $n$-bus distribution system, typically containing only a slack bus and load buses, the AC power-flow problem determines bus voltage magnitudes and phase angles from specified injections based on a coupled system of $2(n-1)$ nonlinear equations. Classical solvers can be accurate but scale poorly and may fail to converge under realistic operating conditions. Simplified linear methods are computationally efficient but unreliable for distribution grids, where higher resistance-to-reactance (R/X) ratios cause more voltage drops, which can lead to greater active power losses and impact voltage stability. These limitations have motivated increasing attention to machine learning approaches, which approximate the mapping from loads and network parameters to voltages without repeated numerical iterations. Recent studies, particularly those using Graph Neural Networks (GNNs), demonstrate that ML-based solvers can achieve orders-of-magnitude speedups while retaining strong accuracy and flexibility (Donon et al., 2020; Jeddi & Shafieezadeh, 2021). Unlike tabular models that must be retrained for each topology, GNNs naturally adapt to varying grid sizes and configurations with little structural modification. This adaptability makes them well-suited for distribution systems, which are characterized by heterogeneity and frequent reconfigurations. However, purely data-driven models suffer from poor generalization, especially under unseen configurations or operating conditions, a key barrier for deployment in real grids. This motivates a systematic investigation into how inductive biases can guide ML models toward more accurate, generalizable, and reliable predictions.

Inductive biases are assumptions embedded in a model's architecture, training process, or data representation that guide learning toward specific solutions. For voltage prediction, relevant inductive biases include the integration of graph-based models, physical laws, or operational constraints. These biases help models better capture the underlying structure of the problem, leading to improved performance and robustness. Among inductive biases, physics-informed approaches have received particular attention as they embed prior knowledge of the system's governing equations to enforce physical consistency in a supervised or unsupervised training regime. Nevertheless, most prior research has focused narrowly on equation-constrained losses, without investigating alternative ways

of embedding physical knowledge. Moreover, little is known about the empirical impact of these inductive biases on generalization performance, which is crucial for deployment in real-world grids where unseen configurations are the norm.

In this work, we investigate the role of inductive biases in learning voltage prediction for distribution grids. Specifically, we analyze three classes of physics-informed inductive biases:

1. **Power-flow-constrained loss functions**, which incorporate the physical equations directly into the optimization objective.

2. **Complex-valued neural networks**, which align model representations with the natural complex-valued structure of electrical quantities.

3. **Task reframing as residual prediction**, where the model learns to approximate deviations from a baseline solver rather than absolute voltages.

For each inductive bias, controlled experiments are conducted across a dataset of common distribution grid topologies, evaluating standard predictive accuracy as well as generalization performance under unseen network configurations [1]. Due to their inherent generalization advantage, we focus on Graph Neural Network architectures in this study. By isolating and comparing the presented inductive biases, we provide a systematic assessment of their effectiveness in guiding learning toward physically meaningful and generalizable solutions, highlighting new pathways for learning voltage prediction. Ultimately, our findings provide concrete guidance for the design of ML models that are not only fast and accurate but also reliable for safety-critical, real-world deployment in modern distribution grids. In this light, we make the following contributions:

- We provide a systematic assessment of standard accuracy and out-of-distribution generalization of physics-informed inductive biases for voltage prediction across heterogeneous distribution networks.

- We contribute the first complex-valued neural network architecture for distribution grid voltage prediction and demonstrate its generalization advantage.

- We derive actionable design guidelines, identifying which methods most effectively enhance real-world reliability and where more research is required.

## 2 LITERATURE REVIEW

### 2.1 ROLE OF INDUCTIVE BIASES IN MACHINE LEARNING

Inductive biases are the foundational assumptions embedded in a machine learning model or algorithm. They encode prior knowledge, constraints, or assumptions to guide how a model generalizes from training data to unseen examples. Thus, the design and selection of inductive biases allow models to perform well in varied settings, boosting generalization and interpretability, while also imposing tradeoffs between efficiency and flexibility. These inductive biases are essential when learning from finite data, as they help constrain the hypothesis space and enable effective learning. Without assumptions, there are infinitely many functions that can fit the training data equally well, so machine learning models need a way to prioritize certain hypotheses over others.

Inductive biases have been systematically applied via algorithmic design (i.e., neural network architecture), training strategies (e.g., regularization), priors in Bayesian inference, and even feature selection. For example, one of the simplest inductive biases often followed in machine learning is Occam's razor (Blumer et al., 1987), which suggests the preference for simpler functions over more complex ones. Occam's razor is often implemented via regularization techniques that penalize model complexity, such as L1 or L2 regularization, and is one of the first principles taught in machine learning to overcome overfitting. However, inductive biases can take many other forms, such as the smoothness bias in kernel methods (Schölkopf & Smola, 2002), sparsity bias in Lasso regression (Tibshirani, 1996), maximum margin in SVMs (Cortes & Vapnik, 1995), and locality assumptions in nearest neighbors algorithms (Cover & Hart, 1967). In deep learning, architectural choices such as convolution or recurrent layers impose structural priors, often enabling remarkable

---

[1]All experiment models and code available at: [Under review]

generalization without explicit specification of inductive biases. When the assumptions align with the structure of the domain, certain biases can accelerate model training or make algorithms more sample-efficient; without them, a model may simply memorize the training data (overfit) or fail to learn altogether. However, inappropriate bias can lead to poor performance if the assumptions do not match the underlying data distribution. This highlights the importance of carefully selecting and designing inductive biases to suit the specific problem and data at hand.

## 2.2 PHYSICS-INFORMED MACHINE LEARNING IN POWER SYSTEMS

Physics-informed (PI) machine learning integrates physical laws and domain knowledge into machine learning models to enhance their performance, interpretability, and reliability (Karniadakis et al., 2021). This approach is particularly relevant in engineering domains like power systems, where physical principles govern system behavior. By embedding these principles into the learning process, this technique aims to provide more trustworthy models that can generalize better to unseen scenarios. In power systems, physics-informed neural networks (PINN) have been applied to various tasks, including state estimation, dynamics analysis, and optimal power flow (Huang & Wang, 2023). Although these approaches claim to improve model generalization, there have been no dedicated generalization studies of these methods for power systems, thus motivating our systematic assessment. In the following, we review three strategies of physics-informed machine learning that are particularly relevant to our study of voltage prediction in distribution grids.

### 2.2.1 POWER-FLOW-CONSTRAINED LOSS FUNCTION

Constraining loss functions is one of the most common approaches to physics-informed machine learning in power systems and was highlighted as a core PINN paradigm by a recent review paper (Huang & Wang, 2023). This is typically achieved by assigning physical meanings to the variables in the output neural networks (NNs) and embedding the governing equations of these physical variables as an additional loss term. The loss function can thus be formally written as:

$$\mathcal{L} = L_{\text{pred}}(\hat{\mathbf{y}}, \mathbf{y}) + \lambda L_{\text{reg}}(\mathbf{W}) + \gamma L_{\text{phys}}(\mathbf{X}, \hat{\mathbf{y}})$$

where $L_{\text{pred}}$ is a typical loss function (such as mean-squared error) which measures the difference between the predicted and target output ($\hat{y}$ and $y$ respectively); $L_{\text{reg}}$ is a regularization term (such as L1 or L2 norm) imposed on the weights $W$ of the neural network; and $L_{\text{phys}}$ is a physical regularization term that quantifies the degree to which the model's predictions violate the governing physical equations, with $X$ representing the input features. $\lambda$ and $\gamma$ are thus hyperparameters that balance the contributions of the different loss components.

For power flow estimation, $L_{\text{phys}}$ is generally composed of the power balance equations of the AC power flow formulation, or a linearized version of this. More specifically, using $P$ as the active power, $Q$ as the reactive power, $V$ as the voltage magnitude, and $\theta$ as the voltage angle, we can express the AC power flow loss term as follows:

$$L_{P_i} = P_i - \sum_{k=1}^{N} |V_i||V_k| \left( G_{ik} \cos(\theta_i - \theta_k) + B_{ik} \sin(\theta_i - \theta_k) \right)$$

$$L_{Q_i} = Q_i - \sum_{k=1}^{N} |V_i||V_k| \left( G_{ik} \sin(\theta_i - \theta_k) - B_{ik} \cos(\theta_i - \theta_k) \right)$$

$$L_{\text{phys}} = \sum_{i=1}^{N} (L_{P_i}^2 + L_{Q_i}^2)$$

where $i$ is a particular bus in an $N$-bus network, and $G_{ik}$ and $B_{ik}$ are the real and imaginary parts of the admittance between buses $i$ and $k$, respectively.

Since this method applies to the training process itself, it is model-agnostic and can be adapted for arbitrary neural network architectures. Due to the ubiquity of this approach, we only highlight the

most relevant studies here. GraphNeuralSolver (Donon et al., 2020) was one of the first works to apply this technique for power flow estimation, using their own custom GNN architecture and serving as a benchmark for many future works in this direction. (Jeddi & Shafieezadeh, 2021) integrates this method into a Graph Attention Network (GAT) model in order to learn the importance of neighbouring nodes, while (Böttcher et al., 2023) adopts a randomized, recurrent GNN architecture, so that its predictor learns how to solve the power flow problem from many different starting points. All three works solve the problem in a strictly unsupervised manner, completely skipping the need for labeled training data and the $L_{\text{pred}}$ term. Other notable works include (Zamzam & Sidiropoulos, 2020) and (Habib et al., 2024) who successfully apply this technique to distribution grid state estimation using non-GNN architectures. However, many of these works focus their analysis on the transmission system context, and none of them evaluate the generalization performance across a structured suite of distribution grid configurations.

### 2.2.2 COMPLEX-VALUED NEURAL NETWORKS

Complex-Valued Neural Networks (CVNNs) extend traditional real-valued neural networks by allowing weights, biases, and activations to be complex numbers. This enables the model to capture phase relationships and interactions between real and imaginary components more naturally, potentially leading to improved performance in tasks involving complex-valued data. CVNNs have been shown to be effective in various domains, including computer vision, speech and signal processing, medical image reconstruction, and control systems (Lee et al., 2022).

A typical layer computes

$$z' = Wz + b, \quad W \in \mathbb{C}^{m \times n}, \ z \in \mathbb{C}^n,$$

which in real–imaginary form expands as

$$z' = (W_r x - W_i y) + i(W_r y + W_i x), \quad (z = x + iy).$$

Nonlinearities are either applied directly component-wise, $\sigma(z) = \sigma(x) + i\sigma(y)$, or in polar form, $\sigma(z) = f(|z|)e^{ig(\angle z)}$, where $f$ and $g$ are nonlinear functions applied to the magnitude and phase, respectively. Training employs *Wirtinger derivatives*, which calculates partial derivatices based on the complex weights and their complex conjugates, allowing standard gradient-based optimization in $\mathbb{C}$. Normalization and loss functions are defined analogously to the real case, often by acting on real and imaginary parts separately or jointly via $|z|$. For more details on CVNNs, the reader is referred to (Lee et al., 2022). In power systems, CVNNs naturally accommodate phasor quantities such as complex voltages and power flows, preserving their inherent algebraic structure.

To date, there are relatively few works that explore complex-valued neural networks for power system applications, despite the fact that electrical quantities such as voltages and currents are inherently complex-valued. For distribution grid voltage prediction, this allows voltage magnitudes and angles to be modeled jointly, rather than separately as in real-valued models. Though not directly a neural network model, the recently-proposed Physics-Informed Symbolic Regression (PISR) method leverages complex-valued representations to learn analytical approximations of the power flow equations (Eichhorn et al., 2025). The model embeds complex information directly into the symbolic regression process, drastically elevating the performance and allowing the model to learn more reliably with less data samples compared to the MLP baseline. This study is limited to two small-scale systems, but the results are indicative of the promise of complex-valued representations for power system tasks. A much earlier work made a first attempt at a CVNN for load flow prediction and noted improved reliability compared to a real valued neural network baseline (Ceylan et al., 2005). However, both models were very small MLPs, consisting of one hidden layer of maximum 8 neurons, and evaluation was performed on a toy 6-bus network. Thus, the results seem hardly tractable for a real-world setting. Lastly, a more comprehensive study is presented in (Wu et al., 2023), which introduces a complex-valued spatio-temporal graph convolutional network (Cplx-STGCN) for the task of power system state forecasting (PSSE) and highlights the advantages of this model against several baselines. However, their performance increase cannot be attributed to the complex valued network, as their CplxFNN model performs poorly and they do not evaluate real-valued counterpart of Cplx-STGCN. Similarly, their evaluation is limited to a single IEEE Additionally, the evaluation examines a single IEEE 118-bus transmission system. Furthermore, the PSSE task is fundamentally different from our setting as it aims to predict unknown voltages in a network using historical voltages and a set of available measurements, while our voltage prediction

task aims to estimate complex nodal voltages given only complex power measurements, effectively learning the power balance equations. To our knowledge, this study contributes the first evaluation of complex valued neural networks for voltage prediction in a realistic distribution grid setting and illustrates their efficacy in robust voltage estimation for out-of-distribution use cases.

### 2.2.3 Residual Prediction

Residual prediction reframes the learning task from predicting absolute values to predicting deviations from a known baseline or approximate solution. This can simplify the learning problem, as the model only needs to learn the residuals, which are often smaller in magnitude and may exhibit simpler patterns than the original target variable. The approach has been popularized in models such as ResNet (He et al., 2016) and XGBoost (Chen & Guestrin, 2016), which have had significant impacts in domains like computer vision and time-series forecasting.

Due to the complex physics of power systems, machine learning training can be unstable and heavily reliant on initializations and hyperparameters. Currently, state of the art power-flow methods typically predict the absolute voltage targets directly. However, voltage magnitudes and angles are generally close to a constant baseline set by the slack bus, thus motivating the use of residuals. By focusing on capturing deviations, the redundant learning of the trivial baseline is eliminated, thereby reducing complexity. Residual learning was employed in (Chen & Zhang, 2023) for probabilistic power flow and simulated on standard transmission system test cases. Beyond voltage prediction, (Eddin Za'ter et al., 2025) use residual learning to learn the nonlinear residual between a given DC-OPF solution and the corresponding AC-OPF solution. When compared against models that directly learn the AC-OPF solution, the residual-learning framework achieved a 35–45% reduction in voltage, power, and feasibility errors, demonstrating its effectiveness for OPF prediction. The success of this approach suggests that residual learning may also be advantageous for voltage prediction.

## 3 Methodology

### 3.1 Experiment Setup

The experimental overview is presented in Figure 1. We evaluate a baseline model and physics-informed variants in the tasks of in-distribution and out-of-distribution voltage prediction. For controlled comparison of the three inductive biases, we conduct a series of experiments using a consistent dataset, model architecture, and evaluation metrics. The ENGAGE dataset is a collection of low and medium voltage grids based on the SimBench networks. This test suite was introduced by Okoyomon & Goebel (2025) to evaluate generalization capabilities of power flow models across several distribution grid configurations and promote robust grid planning and operation. With 3000 sample networks derived from 10 base distribution grids, the dataset contains test cases with varying sizes, topologies, and electrical characteristics, making it ideal for evaluating generalization performance. A summary of the experiment data and task formulation can be found in Appendix C.

We first establish a baseline to identify how well each model performs under known network configurations and learned data distributions. We randomly shuffle the data with all grids and adopt a 75/15/10 train/validation/test split for model training. To evaluate generalization performance, we conduct an out-of-distribution (OOD) experiment. We use a leave-one-out approach where we train on all but one of the base grid configurations and test on the held-out configuration. We thus apply the same train/validation/test split, with 300 samples (10%) in the held-out configuration. This approach allows us to assess how well each model can generalize to unseen grid topologies and operating conditions, which is critical for real-world deployment. Though a one-to-one grid evaluation is possible, previous studies indicate that current ML solvers are not mature enough for this use case (Böttcher et al., 2023; Hansen et al., 2022; Jeddi & Shafieezadeh, 2021; Okoyomon & Goebel, 2025). As such, we focus on the OOD setting for our generalization study.

To evaluate model performance, we employ the standard metric of Root Mean Squared Error (RMSE) between the predicted and true voltage magnitudes and angles across all buses in the network. Additionally, we report the training time, inference time, and model capacity to assess computational efficiency. To evaluate generalization performance, the RMSE of each held-out configuration is compared against the RMSE on the baseline test set. A tighter and lower RMSE distribution on the held-out configuration indicates better generalization.

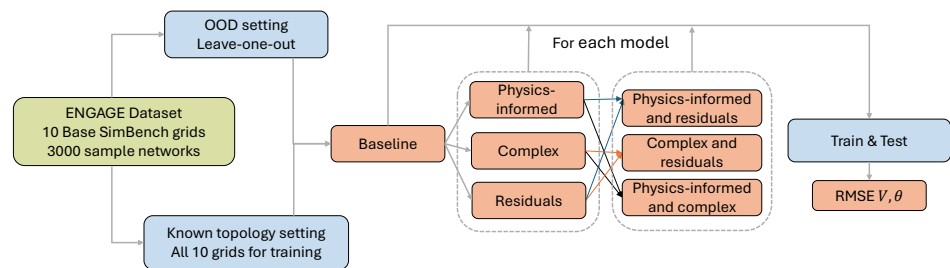

Figure 1: Experiment methodology for in-distribution and out-of-distribution model evaluation

## 3.2 MODEL DESIGN

We provide a baseline GNN architecture to serve as a control model for our experiments. We employ a GraphConv-based message-passing architecture with batch normalization, following the standard GNN framework established by Morris et al. (2019). This provides strong representational capacity while maintaining theoretical grounding in GNN expressivity research. As input, the model accepts the active and reactive power injections and topological distance to slack bus as node features, and the line resistances and reactances as edge features. Additionally, every graph has access to the slack bus information (slack voltage magnitude, angle, resistance, and reactance) as global features. The model's task is then to predict the voltage magnitude and angle at every bus.

The model utilizes two-stage preprocessing and postprocessing layers, implemented using fully connected networks with ReLU activations and batch normalization. The core processing consists of 7 128-dimensional message-passing layers using the GraphConv update scheme. To attribute generalization performance to inductive bias rather than the specific GNN backbone, we validate results through additional experimentation on a GAT backbone (Brody et al., 2021) in Appendix B.2.

For all models, we keep the architecture as similar as possible to the baseline, only changing the components necessary to implement each inductive bias. This allows us to isolate the effects of each inductive bias while minimizing confounding factors. As an additional analysis, we present results of combined inductive bias models in Appendix B.1. For all models, we apply uniform training procedures to ensure a fair comparison, using MSE as the supervised loss function, 3000 epochs, batch sizes of 128, and an Adam optimizer with an adaptive learning rate starting from $10^{-3}$ until $10^{-5}$. Using a hyperparameter tuning, we determine the optimal model configuration and ensure strong performance on standard accuracy metrics before proceeding with the experiment variants.

## 3.3 POWER-FLOW-CONSTRAINED LOSS MODEL

We implement a loss function that combines the standard supervised loss with a physics-based regularization that penalizes violations of the AC power flow equations. More specifically, we train the baseline GNN as usual, and then fine-tune the model using the physics informed loss for 20 epochs. This limit was deliberately chosen to ensure this phase acts solely as a structural refinement, preventing it from becoming a deep re-optimization of the landscape. We observed no significant gain in OOD performance when extending this phase up to 30 epochs. This scheme proved to be much more reliable than training completely unsupervised or with a weighted physics loss, since these methods are highly reliant on hyperparameters. Furthermore, since the data is derived using pandapower (Thurner et al., 2018), we replicate their AC power balance equations and model assumptions to implement our loss function. We ensure that the physics informed loss function provides near-zero values for the ground truth voltage targets before using it for model training.

## 3.4 COMPLEX-VALUED MODEL

We implement a GNN architecture that operates on complex-valued inputs and outputs, using complex-valued layers and activation functions. This approach allows the model to naturally capture the relationships between voltage magnitudes and angles, which are inherently complex-valued quantities in power systems. Where applicable, we use the complexPyTorch library (Popoff, 2021)

Table 1: Model Performance Comparison: OOD Experiments Statistical Summary

| Model | RMSE VM (p.u.) | | | | RMSE VA (deg) | | | |
|--------|--------|--------|--------|--------|--------|--------|--------|--------|
| | **Min** | **Max** | **Mean** | **Std** | **Min** | **Max** | **Mean** | **Std** |
| **Base** | 0.0087 | 0.0287 | 0.0179 | 0.0062 | 0.6049 | 2.7343 | 1.3271 | 0.6458 |
| **Res** | 0.0094 | 0.0389 | 0.0183 | 0.0083 | 0.6101 | 4.2366 | 1.3419 | 1.0598 |
| **Cplx** | 0.0102 | 0.5842 | 0.0715 | 0.1802 | **0.0093** | **0.0651** | **0.0212** | **0.0179** |
| **PFLoss** | **0.0070** | **0.0266** | **0.0166** | **0.0060** | 0.6013 | 2.3928 | 1.2628 | 0.5595 |

and implement custom complex-valued GNN layers such as complex batch normalization and complex GraphConv to ensure compatibility with the base GNN framework.

## 3.5 RESIDUALS MODEL

We implement a GNN architecture that predicts the residuals as deviations from the slack bus voltage. This simplifies the learning task, as the model only needs to learn the differences from a known baseline (the slack bus voltage), which are often smaller in magnitude and may exhibit simpler patterns than the original target variable. We implement the residuals model by enforcing a single skip connection that connects the neural network's output to the slack bus reference value (Donti et al., 2017). This way, the model learns to output residuals, which are added to the slack bus reference values to produce the final voltage predictions. This allows us to keep the input data format consistent across all models, simplifying data handling and preprocessing.

## 4 RESULTS

We evaluate three inductive-biases for voltage prediction in distribution grids compared to a purely data-driven baseline model. We report performance for prediction on grids with familiar structures and on unseen (i.e. OOD) grids. In this section, we present general model performance results using predictive performance and model efficiency metrics. For a complete view of individual benchmarking results and additional experimentation, we refer the reader to Appendices A and B.

Table 1 provides a statistical summary of the generalization experiments, while Figure 2 displays overall performance metrics. Voltage magnitude RMSE is presented using p.u. values, meaning that results are normalized with respect to the reference voltage of the network, allowing for interpretation as relative errors (e.g., 0.01 p.u. is equivalent to 1% of the nominal voltage).

In terms of voltage magnitude, the supervised physics-informed variant achieves the highest reliability across the simulated topologies, indicating greater robustness as a magnitude predictor. The complex-valued model attains the best angle accuracy, with an improvement of two orders of magnitude across all experiments, at the cost of extra computational burden. Although the residual learning model performs the best in certain cases, it does not provide any significant advantages.

## 4.1 BASELINE MODEL

When predicting voltage magnitude on known grid configurations, the baseline GNN attains an average RMSE of $\mathbf{0.0067}$ p.u. for voltage magnitude, and $\mathbf{0.1573}°$ for voltage angle. For the OOD experiments, the baseline yields a mean performance of $\overline{\mathrm{RMSE}}(|V|) = \mathbf{0.0179}$ p.u. and $\overline{\mathrm{RMSE}}(\angle V) = \mathbf{1.3271}°$. Notably, this model proves to be quite stable in its OOD prediction of voltage magnitude. The model contains 302.7K parameters and its training and inference times are approximately $\mathbf{1200}$s and $\mathbf{2}$ms for all experiments.

## 4.2 COMPLEX-VALUED LEARNING

The complex-valued model excels on both magnitude and angle prediction across test cases. It achieves $\mathbf{0.00239}°$ angle RMSE and $\mathbf{0.00555}$ p.u. magnitude RMSE when predicting on known grid configurations, proving to be the best among all methods. Across the generalization experiments, we observe a very tight distribution for both voltage magnitude and angle. One bad per-

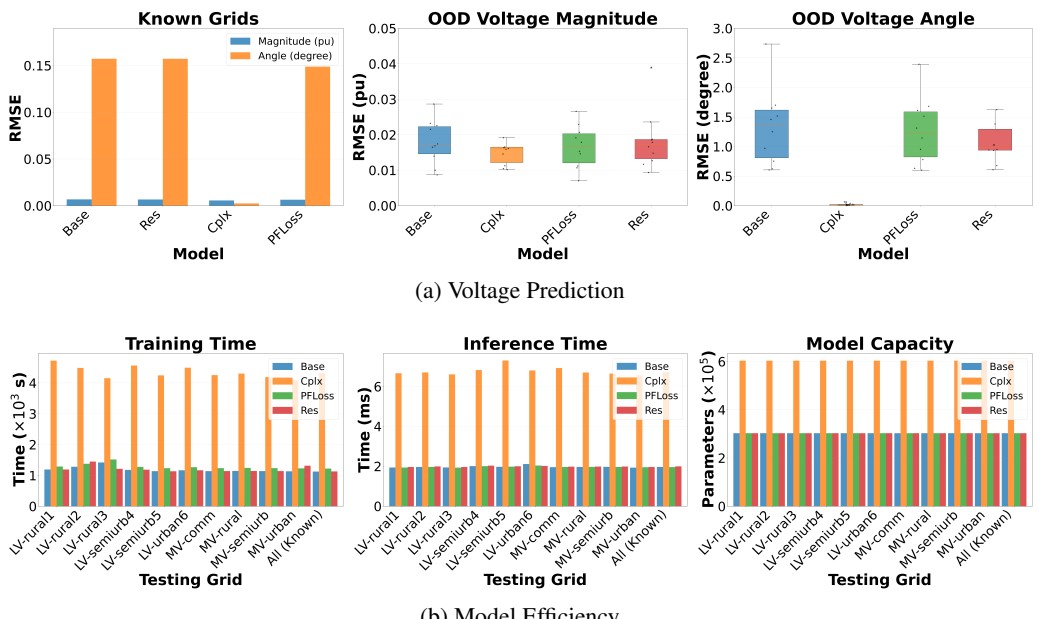

(a) Voltage Prediction

(b) Model Efficiency

Figure 2: Summary of physics-informed model benchmarking metrics across all experiments

formance outlier drives the $\overline{\text{RMSE}}(|V|)$ up to **0.0715** p.u, while $\overline{\text{RMSE}}(\angle V)$ remains stable at **0.0212°**. These results underscore the remarkable angle predictive performance of this model for both familiar and unseen networks. This performance gain comes with the highest computational burden, producing an average of approximately 4x training time, 3x inference time, and 2x model capacity compared to the other variants.

### 4.3 PHYSICS-INFORMED LEARNING

The supervised physics-informed variant exhibits an increase in performance compared to the baseline model for almost all experiments, illustrating the impact of this PI fine-tuning mechanism. For the known grids setting, it reaches $\text{RMSE}(|V|) = \mathbf{0.00638}$ p.u. and $\text{RMSE}(\angle V) = \mathbf{0.1488°}$. Across OOD benchmarks, the method exhibits the most accurate and stable performance for voltage magnitude prediction, with $\overline{\text{RMSE}}(|V|) = \mathbf{0.0166}$ p.u., and the smallest standard deviation. When considering voltage angle, its mean performance ($\overline{\text{RMSE}}(\angle V) = \mathbf{1.263°}$) is only outperformed by the complex-valued model. Model efficiency metrics are very similar to the baseline model, with a slight increase in training time ($\approx 100s$) due to the physics-informed fine-tuning step.

### 4.4 RESIDUAL LEARNING

The residual approach provides performance comparable to the baseline model. When predicting on familiar grids, it attains $\text{RMSE}(|V|) = \mathbf{0.00662}$ p.u. and $\text{RMSE}(\angle V) = \mathbf{0.1573°}$, showing nearly identical performance to the baseline. Across generalization experiments, accuracy and stability are also similar to the baseline, with $\overline{\text{RMSE}}(|V|) = \mathbf{0.01828}$ p.u. and $\overline{\text{RMSE}}(\angle V) = \mathbf{1.342°}$. However, it is worth noting that, unlike the baseline, the residuals model proved to be the best performing model in certain situations, as made evident in the LV-semiurb5 and LV-urban6 OOD experiments. Model efficiencies are comparable to the baseline.

## 5 DISCUSSION

### 5.1 INDUCTIVE BIASES FOR VOLTAGE PREDICTION

When estimating voltage magnitude and angle for networks similar to those seen in training, our new complex-valued model provides the most accurate predictions. Voltage angle is particularly

impressive, significantly outperforming the other variants by approximately two orders of magnitude. Generally, all physics-informed inductive biases seem to improve the results of the already-performant baseline model for the known grids experiment case. This demonstrates that all three inductive biases can be used to assist in data-driven learning of power grid characteristics.

When looking at the results of the generalization experiments, the presence of unseen grid topologies clearly influences model accuracy, with average performance dropping by one order of magnitude on average. In this setting, the physics-informed variant attains the best performance on average for voltage magnitude and the lowest variability across individual OOD grids, though the improvements are minor. The complex-valued model continues to thrive in voltage angle prediction. In fact, the complex-valued model's results on the OOD tests greatly surpass even the predictions of other models under known grid configurations, highlighting its fidelity for voltage angle prediction.

In light of these results, we propose that machine learning models leverage complex-valued neural networks to truly capture the interdependence between the real and imaginary components of the voltage phasors. *Fine-tuning* using physics-informed losses can also be a worthwhile add-on to enhance model accuracy, so long as one has accurate physical equations and relatively accurate initial predictive performance.

## 5.2 REAL-WORLD APPLICATION

The complex-valued variant demonstrates noteworthy performance overall, achieving both high accuracy and stability across most tasks. This model surpasses the ML-based power-flow implementations of previous comparable studies, particularly in voltage angle prediction (Lin et al., 2024; Suri & Mangal, 2025; Okhuegbe et al., 2024; Hu et al., 2020). Contrary to prevalent data-driven approaches, the complex-valued variant naturally captures the inherent coupling between voltage magnitude and angle, treating them as interdependent variables. This precision helps satisfy a critical requirement in distribution system voltage prediction, where real power flows depend on angle differences and incorrectly estimated angles can lead to instability, congestion, or cascading failures.

The proposed physics-informed variant demonstrates particularly robust performance in predicting voltage magnitudes across diverse grid topologies. The incorporation of the AC power flow equations as a regularization term leads the learning process to remain consistent with Kirchhoff's laws, thereby reducing the likelihood of systematic deviations that might exist otherwise when extrapolating to unseen grid configurations.

Over known topologies, all models achieve a voltage magnitude RMSE of roughly $0.5\%$, matching the accuracy of iMSys household meters (EFR GmbH, 2022) and aligning with the practical noise floor of commercially deployed measurement hardware. Voltage angle errors are also well below the $1°$ metering standard, reaching RMSE values up to three orders of magnitude below this threshold. These findings support the use of such models for distribution grid operation in familiar networks, as they can safely be deployed with limited uncertainty. However, under the OOD testing, voltage angle errors increase by one order of magnitude and only the complex-valued model reliably stays under this $1°$ threshold over all cases. A similar observation can be made for voltage magnitude prediction, as even the best performing model produces errors that are higher than the 0.5% measurement standard by one order of magnitude. This highlights the significance of comprehensive generalization studies for robust assessment of machine learning in distribution grids and represents a key finding for establishing practically viable solutions for real-world deployment. Unlike previous works that primarily assess performance on seen or similarly sized topologies, we systematically show that state of the art GNN architectures do not attain the necessary level of accuracy in OOD scenarios. While physics-informed inductive biases may improve the generalization capability, only for the complex valued model did performance stay below the desired threshold for voltage angle, while all other variants exceed acceptable error margins. This indicates the need for a shift in focus in machine learning methodology from standard predictive accuracy to generalization performance assessment in order to take meaningful steps towards real-world application.

As an additional interpretation, we compare the best performing models for voltage prediction to a commonly used simplification of the full AC Power Flow solution: DC Power Flow. The DC power flow approximation linearizes the full AC method such that a non-iterative, convergent solution can be found. However, this simplification ignores reactive power and assumes small angle differences and a fixed voltage magnitude of 1.0 p.u. for all buses, and thus often leads to inaccurate predictions

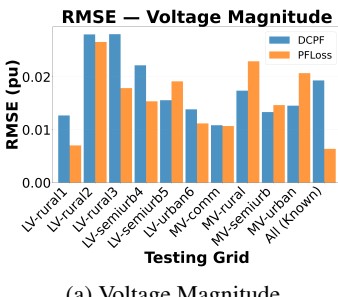
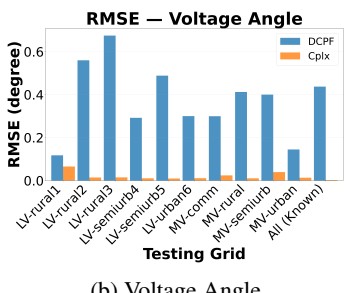

(a) Voltage Magnitude

(b) Voltage Angle

Figure 3: Comparison of Best-Performing models to DC PF

in situations with larger voltage drops. We present this baseline not due to its sophistication but rather due to its relevance as an empirically observed baseline for generalization failure, as established in prior benchmarking studies (Hansen et al., 2022; Yaniv et al., 2023; Okoyomon & Goebel, 2025). These works commonly observe the limitation that GNN-based power flow methods perform well on known-configurations but are ineffective in predicting PF from unseen grids during training, and are usually outperformed by simplified models such as DC PF. Looking at Figure 3, we see that our PFLoss GNN solver is able to significantly outperform DCPF for known networks and match its performance for unseen grids. Additionally, by using complex-valued neural networks, we are able to outperform DC PF by one order of magnitude in voltage angle prediction across all test cases. These results emphasize the promise of physics-informed models as a robust replacement for existing analytical models. For a secondary analytical baseline, we compare performance results to the well-established LinDistFlow method (Baran & Wu, 1989) in Appendix D.2.

## 5.3 CONCLUSION AND FUTURE WORK

This study evaluates three inductive-bias strategies for voltage prediction in distribution grids. Using a dataset of heterogeneous low- and medium-voltage networks, we assess performance across familiar topologies and out-of-distribution cases. Our findings highlight complementary strengths. Our novel complex-valued model achieves state-of-the-art angle prediction for both known and unknown grids, surpassing all other approximations, though at higher computational cost. The physics-informed variant consistently improves magnitude prediction and exhibits the lowest variability across unseen topologies, demonstrating robustness rooted in physical consistency, but this is only a slight improvement above the strong baseline. In contrast, the residual learning approach provided limited overall gains, though it excelled in select cases and seems to be able to be synergetic with other methods. Collectively, these results show that carefully chosen inductive biases substantially enhance the reliability of machine learning models for power system analysis.

Future research should extend these insights along three directions. Firstly, a deeper examination of physics-constrained training regimes is needed, including theoretical analysis and open benchmarking, to clarify when physics-informed losses improve generalization. For our particular study, this variant was the most sensitive to hyperparameters and required significant knowledge of the pandapower modeling library to produce an effective solution. Even then, the physics-informed loss could not be used competitively in a purely unsupervised manor, with performance results two orders of magnitude worse than the other variants at best. This finding comes at no surprise when considering the non-linear, non-convex nature of the AC power balance equations, but is nonetheless in conflict with several other works that tout its success. Secondly, evaluation on non-European grid datasets (e.g. Smart-DS) would test the applicability of these methods in unbalanced, three-phase networks with different structural characteristics. Finally, exploring hybrid models that combine multiple inductive biases could yield situation-dependent predictors, though early results suggest that such benefits are non-additive and grid-specific (Appendix B.1).

By systematically studying inductive biases, this work contributes to the development of machine learning models that are not only accurate but also robust enough for deployment in real-world power system operations. Ultimately, our results indicate that physics-informed and complex-valued approaches can move ML-based solvers beyond academic benchmarks toward practical tools for reliable voltage prediction in distribution grids.

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

# A COMPLETE BENCHMARKING RESULTS

We present the full benchmarking results in Table 2, which provides the raw performance metrics for all inductive bias models across all testing grids, using RMSE, training time, inference time, and model capacity as evaluation criteria.

Table 2: Raw Model Performance Results by Testing Grid

| Testing Grid | Model | RMSE VM | RMSE VA | Train (s) | Inference (ms) | Capacity |
|---|---|---|---|---|---|---|
| LV-rural1 | Base | 0.00868 | 2.73428 | **1191.9** | 1.93844 | 302.7K |
| LV-rural1 | Cplx | 0.58421 | **0.06514** | 4722.3 | 6.66419 | 603.4K |
| LV-rural1 | PFLoss | **0.00703** | 2.39278 | 1287.7 | **1.93717** | 302.7K |
| LV-rural1 | Res | 0.03893 | 4.23657 | 1192.3 | 1.96439 | 302.7K |
| LV-rural2 | Base | 0.02866 | 1.70101 | **1280.7** | **1.96457** | 302.7K |
| LV-rural2 | Cplx | **0.01459** | **0.01416** | 4486.7 | 6.70308 | 603.4K |
| LV-rural2 | PFLoss | 0.02658 | 1.68133 | 1376.7 | 1.96591 | 302.7K |
| LV-rural2 | Res | 0.01786 | 1.38314 | 1447.8 | 1.99109 | 302.7K |
| LV-rural3 | Base | 0.01650 | 0.97081 | 1419.6 | 1.94195 | 302.7K |
| LV-rural3 | Cplx | **0.01612** | **0.01459** | 4153.4 | 6.60584 | 603.4K |
| LV-rural3 | PFLoss | 0.01787 | 0.95436 | 1517.4 | **1.93280** | 302.7K |
| LV-rural3 | Res | 0.01662 | 0.93536 | **1213.8** | 1.96289 | 302.7K |
| LV-semiurb4 | Base | 0.01690 | 0.75439 | **1179.3** | 2.01071 | 302.7K |
| LV-semiurb4 | Cplx | 0.01657 | **0.01053** | 4564.2 | 6.82323 | 603.4K |
| LV-semiurb4 | PFLoss | **0.01536** | 0.78076 | 1275.9 | **2.00547** | 302.7K |
| LV-semiurb4 | Res | 0.01854 | 0.95156 | 1186.3 | 2.03135 | 302.7K |
| LV-semiurb5 | Base | 0.02154 | 0.60486 | 1137.9 | **1.97208** | 302.7K |
| LV-semiurb5 | Cplx | 0.01590 | **0.00933** | 4244.1 | 7.29975 | 603.4K |
| LV-semiurb5 | PFLoss | 0.01914 | 0.60126 | 1233.5 | 1.98107 | 302.7K |
| LV-semiurb5 | Res | **0.01481** | 0.61009 | **1134.1** | 2.00147 | 302.7K |
| LV-urban6 | Base | 0.01000 | 0.62547 | 1165.8 | 2.11327 | 302.7K |
| LV-urban6 | Cplx | 0.01129 | **0.01110** | 4493.8 | 6.80414 | 603.4K |
| LV-urban6 | PFLoss | 0.01118 | 0.63158 | 1265.2 | 2.03692 | 302.7K |
| LV-urban6 | Res | **0.00937** | 0.67909 | **1164.8** | **2.01710** | 302.7K |
| MV-comm | Base | 0.01396 | 1.25189 | **1138.5** | **1.95523** | 302.7K |
| MV-comm | Cplx | **0.01045** | 0.02395 | 4253.3 | 6.92224 | 603.4K |
| MV-comm | PFLoss | 0.01068 | 1.14607 | 1234.6 | 1.96308 | 302.7K |
| MV-comm | Res | 0.01269 | 1.02931 | 1138.9 | 1.98173 | 302.7K |
| MV-rural | Base | 0.02256 | 1.45830 | 1145.9 | **1.96252** | 302.7K |
| MV-rural | Cplx | **0.01637** | **0.01055** | 4303.7 | 6.70011 | 603.4K |
| MV-rural | PFLoss | 0.02293 | 1.31355 | 1242.5 | 1.96917 | 302.7K |
| MV-rural | Res | 0.02361 | 0.94275 | **1143.2** | 1.98348 | 302.7K |
| MV-semiurb | Base | 0.01742 | 1.65200 | **1138.9** | 1.97044 | 302.7K |
| MV-semiurb | Cplx | 0.01917 | **0.03934** | 4189.4 | 6.64684 | 603.4K |
| MV-semiurb | PFLoss | **0.01465** | 1.61223 | 1235.0 | **1.96650** | 302.7K |
| MV-semiurb | Res | 0.01872 | 1.62399 | 1139.8 | 1.98686 | 302.7K |
| MV-urban | Base | 0.02322 | 1.51757 | **1131.7** | **1.93205** | 302.7K |
| MV-urban | Cplx | **0.01017** | 0.01314 | 4083.5 | 6.55564 | 603.4K |
| MV-urban | PFLoss | 0.02069 | 1.51453 | 1227.6 | 1.95075 | 302.7K |
| MV-urban | Res | 0.01166 | 1.02719 | 1313.9 | 1.96254 | 302.7K |
| All (Known) | Base | 0.00674 | 0.15726 | **1125.2** | 1.96604 | 302.7K |
| All (Known) | Cplx | **0.00555** | **0.00239** | 4311.9 | 6.71587 | 603.4K |
| All (Known) | PFLoss | 0.00638 | 0.14876 | 1221.8 | **1.95872** | 302.7K |
| All (Known) | Res | 0.00662 | 0.15726 | 1128.9 | 1.99554 | 302.7K |

# B ADDITIONAL EXPERIMENTS

## B.1 HYBRID MODELS

The previously-introduced physics-informed models only consider one variant per inductive bias, which naturally leaves the question of whether combining them may result in improved performance. As a result, we implement and evaluate all pairwise combinations of our physics-informed variants to create physics-informed hybrid models. Namely, we present three novel hybrids: *Cplx + PFLoss*, *Cplx + Res*, and *PFLoss + Res*. To our knowledge, no previous study has introduced such models for the task of distribution grid voltage prediction.

Table 3: OOD Performance of Hybrid vs Non-Hybrid Inductive Bias Models

| Model | RMSE VM (p.u.) | | | | RMSE VA (deg) | | | |
| --- | --- | --- | --- | --- | --- | --- | --- | --- |
| | Min | Max | Mean | Std | Min | Max | Mean | Std |
| **Base** | 0.0087 | 0.0287 | 0.0174 | 0.0063 | 0.6049 | 2.7343 | 1.3059 | 0.6813 |
| **Res** | 0.0094 | 0.0389 | 0.0190 | 0.0085 | 0.6101 | 4.2366 | 1.3769 | 1.1180 |
| **Cplx** | 0.0104 | 0.5842 | 0.0783 | 0.1897 | **0.0093** | 0.0651 | 0.0221 | 0.0188 |
| **PFLoss** | **0.0070** | 0.0266 | 0.0166 | 0.0060 | 0.6013 | 2.3928 | 1.2628 | 0.5595 |
| **Cplx + PFLoss** | 0.0095 | 0.1450 | 0.0342 | 0.0414 | 0.0098 | 0.2996 | 0.0465 | 0.0894 |
| **Cplx + Res** | 0.0083 | 0.0414 | 0.0201 | 0.0110 | 0.0103 | **0.0298** | **0.0188** | **0.0059** |
| **PFLoss + Res** | 0.0093 | **0.0244** | **0.0163** | **0.0049** | 0.6363 | 2.5982 | 1.1516 | 0.6145 |

In Table 3 we compare the original single-inductive bias models (in the top section) to the new hybrid models (in the bottom). The best result in each column is highlighted in **bold** while the second-best is underlined. We observe from the table that augmenting the complex and physics models with the residual-learning technique enhances the performance of each of these models, resulting in the best performance for their respective subtasks (voltage magnitude prediction for Phys-Loss + Residuals and voltage angle prediction for Complex + Residuals). However, combining the previously-best-performing Phys-Loss and Complex models does not yield any added benefit, and results in a model that generally weakens the advantage of each original bias. Ultimately, the Phys-Loss + Residuals and the Complex + Residuals model are the most stable for generalization for voltage magnitude and angle, respectively, but they present no significant advantage over the single-inductive bias models. These results further emphasize the consistent improvement in voltage angle prediction due to complex-valued networks, as all complex-valued model errors are magnitudes lower than the real-valued counterparts.

The full raw results for each hybrid model experiment are presented in Table 9.

## B.2 CROSS-ARCHITECTURE VALIDATION

As the backbone of our main inductive bias models, we use a GraphConv architecture for message passing between nodes. The node update scheme of the GraphConv message passing mechanism is as follows:

$$x_i' = W_1 x_i + W_2 \sum_{j \in N(i)} e_{ij} x_j$$

Where:

- $\mathbf{x}_i'$ is the **output feature vector** for target node $i$ after the GraphConv layer
- $\mathbf{x}_i$ is the **input feature vector** for target node $i$
- $\mathcal{N}(i) \cup \{i\}$ is the set of node $i$'s **neighbors** (including $i$ itself)
- $e_{ij}$ is the **pre-determined edge weighting** of source node $j$'s message to target node $i$
- $\mathbf{W}_1$ is the learnable **target weight matrix** applied to the target node's features, $\mathbf{x}_i$
- $\mathbf{W}_2$ is the learnable **source weight matrix** applied to the aggregated source node features

As cross-architecture ablation study, we re-run the baseline and all single inductive bias models on a GAT backbone, since GAT represents a different class of GNNs (when compared to GraphConv) that uses input-dependent weighting rather than fixed/learned shared weights for all neighbours. The node update scheme of the GAT message passing mechanism is as follows:

$$\mathbf{x}'_i = \sum_{j \in \mathcal{N}(i) \cup \{i\}} \alpha_{ij} \mathbf{W}_s \mathbf{x}_j$$

$$\alpha_{ij} = \frac{\exp\left(\mathbf{a}^\top \mathrm{LeakyReLU}\left(\mathbf{W}_t \mathbf{x}_i + \mathbf{W}_s \mathbf{x}_j\right)\right)}{\sum_{k \in \mathcal{N}(i) \cup \{i\}} \exp\left(\mathbf{a}^\top \mathrm{LeakyReLU}\left(\mathbf{W}_t \mathbf{x}_i + \mathbf{W}_s \mathbf{x}_k\right)\right)}$$

Where:

- $\mathbf{x}'_i$ is the **output feature vector** for target node $i$ after the GATv2 layer
- $\mathbf{x}_i$ is the **input feature vector** for target node $i$
- $\mathcal{N}(i) \cup \{i\}$ is the set of node $i$'s **neighbors** (including $i$ itself)
- $\alpha_{ij}$ is the **normalized attention coefficient** (weight) of source node $j$'s message to target node $i$
- $\mathbf{W}_t$ is the **target linear weight matrix** (learnable parameter) applied to the target node's features, $\mathbf{x}_i$
- $\mathbf{W}_s$ is the **source linear weight matrix** (learnable parameter) applied to the source node's features, $\mathbf{x}_j$
- $\mathbf{a}$ is the **attention weight vector** (learnable parameter) used to compute the attention score

The results in Table 4 confirm the efficacy of these inductive biases. Most notably, the complex-valued GAT similarly outperforms the other variants by one to two orders of magnitude in the task of voltage angle prediction. Although we can observe the same fine-tuning effect of the physics-informed loss, its benefits are comparatively mild in this benchmark, with the residuals model providing the largest benefit for voltage magnitude prediction. As in the original GraphConv benchmark, the physics-informed loss model improves against all non-complex-valued models in the task of voltage angle prediction. Furthermore, the baseline GNN is always outperformed by at least one inductive bias model, with the most significant improvement coming from the complex-valued model. The results confirm that we can observe non-architecturally specific trends, validating that the generalization benefits stem from the biases themselves, not the specific GNN backbone.

Table 4: OOD Performance of GAT-backbone Models

| Model | RMSE VM (p.u.) | | | | RMSE VA (deg) | | | |
|---|---|---|---|---|---|---|---|---|
| | Min | Max | Mean | Std | Min | Max | Mean | Std |
| GAT | 0.0121 | 0.1630 | 0.0341 | 0.0457 | 0.4316 | 3.5972 | 1.3452 | 1.0874 |
| GAT-Cplx | 0.0104 | 22.4246 | 2.2593 | 7.0854 | **0.0097** | **0.6084** | **0.0781** | **0.1866** |
| GAT-PFLoss | **0.0092** | 0.1417 | 0.0368 | 0.0420 | 0.4773 | 3.2362 | 1.2649 | 0.9726 |
| GAT-Res | 0.0098 | **0.0656** | **0.0248** | **0.0193** | 0.7012 | 8.2566 | 2.3707 | 2.4294 |

The full raw results for each GAT model experiment are presented in Table 10.

## B.3 MODEL CAPACITY

Due to the representation of complex values as pairs of real numbers in computing systems, training and deploying complex-valued neural networks comes at an extra computational cost (Section 4.2). Particularly, experiment results show that the number of parameters in such models is roughly double of their real-valued counterparts. To ensure that the observed performance benefits of the complex-valued model is not attributed to its increased model capacity, we present extra experiment results comparing the original baseline and complex valued models with a larger baseline model. The dimension of the baseline model message passing layers was increased by 50%, from 128 to 192,

Table 5: Model Capacity Performance Comparison

| Model | Capacity | RMSE VM (p.u.) | | | | RMSE VA (deg) | | | |
|---|---|---|---|---|---|---|---|---|---|
| | | Min | Max | Mean | Std | Min | Max | Mean | Std |
| **Base** | 302.7K | **0.0087** | **0.0287** | **0.0179** | **0.0062** | 0.6049 | 2.7343 | 1.3271 | 0.6458 |
| **Base-L** | 624.2K | 0.0112 | 0.1208 | 0.0281 | 0.0336 | 0.5568 | 18.7405 | 2.6622 | 5.6551 |
| **Cplx** | 603.4K | 0.0102 | 0.5842 | 0.0715 | 0.1802 | **0.0093** | **0.0651** | **0.0212** | **0.0179** |

resulting in a model capacity of 624.2K and making it more comparable to the 603.4K complex model. The results in Table 5 confirm that the improved predictive accuracy is unlikely due to an increase in model capacity.

## C  EXPERIMENT SETTING

### C.1  DATASET

Table 6: ENGAGE Dataset Statistics

| Grid | Type | Buses | Lines | Voltage (kV) | R/X Ratio | Line Loading | | |
|---|---|---|---|---|---|---|---|---|
| | | | | | | Min (%) | Max (%) | Avg (%) |
| **LV1** | Rural | 15 | 14 | 0.4 | 2.41 | 0.15 | 65.28 | 16.70 |
| **LV2** | Rural | 97 | 96 | 0.4 | 2.55 | 0.07 | 57.69 | 9.93 |
| **LV3** | Rural | 129 | 128 | 0.4 | 2.55 | 0.06 | 92.09 | 17.42 |
| **LV4** | Semi-urban | 44 | 43 | 0.4 | 2.52 | 0.73 | 99.99 | 29.98 |
| **LV5** | Semi-urban | 111 | 110 | 0.4 | 1.58 | 0.01 | 91.10 | 19.56 |
| **LV6** | Urban | 59 | 58 | 0.4 | 1.56 | 0.25 | 99.83 | 22.15 |
| **MV1** | Rural | 97 | 102 | 20.0 | 2.62 | 0.08 | 48.46 | 10.21 |
| **MV2** | Semi-urban | 120 | 127 | 20.0 | 2.78 | 0.01 | 88.82 | 18.53 |
| **MV3** | Urban | 136 | 148 | 10.0 | 1.13 | 0.05 | 99.99 | 15.60 |
| **MV4** | Commercial | 106 | 113 | 20.0 | 1.92 | 0.01 | 99.30 | 13.84 |

As depicted in Table 6, the ENGAGE dataset is a heterogenous distribution grid dataset, with 10 unique base distribution grids varying in size and type (Okoyomon, 2025). All grids are generated by applying Powerdata-gen (Donon, Balthazar, 2022) to the low- and medium-voltage distribution networks from SimBench's scenario 1 (future grid with normal increase of distributed energy resources). Each grid type has 300 test cases with varying network loading conditions. Thus, our "leave-one-out" protocol tests scale generalization, as models trained on pools of smaller grids are tested on larger grids and vice versa. These distribution networks have fundamentally different characteristics than transmission networks, as distribution grids have much lower voltage levels, tend to be more radial in structure, and have much higher R/X ratios (approx. 0.1-0.3 is typical for transmission grids while distribution grids can have 1.0-5.0 or more, leading to large voltage drops).

Each graph in the ENGAGE dataset is represented as a PyTorch Geometric Data object. Since we are dealing with voltage prediction in distribution grids, there are only PQ buses and the slack bus. Additionally, we store the slack information globally and add the minimum hops to the slack bus for every bus as an additional feature. The model task is to use the bus information (*data.x*), network connectivity (*data.edge_index*), line information (*data.edge_attr*), and global information (*data.slack_info*) to predict the output voltages at every bus (*data.y*). To assist the physics-informed loss formulation, we additionally include the underlying PYPOWER case file information (*data.ppci*) that were used to model the networks in pandapower. This allows us to correctly formulate our PyTorch power balance equation implementation, since different modeling frameworks make different network assumptions.

Table 7: PyTorch Geometric Data Object Attributes

| Attribute | Dimension | Description / Features |
|---|---|---|
| x | $(N, 3)$ | Node features. $N$ is the number of nodes/buses. Features: [p_mw, q_mvar, hops_to_slack] |
| edge_index | $(2, 2E)$ | Graph connectivity in COO format. $E$ is the number of lines. Edges are directed. |
| edge_attr | $(2E, 4)$ | Edge features. $E$ is the number of lines. Features: [r_pu, x_pu] |
| slack_info | $(4)$ | Static network-wide parameters. Parameters: [slack_vm_pu, slack_va_degree, slack_r_pu, slack_x_pu] |
| y | $(N, 2)$ | Target values for each node. $N$ is the number of nodes/buses. Labels: [vm_pu, va_degree] |
| ppci | N/A | Internal pypower casefile used by pandapower[2] for power flow (net["_ppc"]["internal"]). |

Table 8: Bus types and their attributed in AC Power Flow

| Bus type | $P$ | $Q$ | $V$ | $\theta$ |
|---|---|---|---|---|
| Slack (ref) | Unknown | Unknown | Given | Given |
| PV (gen) | Given | Unknown | Given | Unknown |
| PQ (load) | Given | Given | Unknown | Unknown |

## C.2 VOLTAGE PREDICTION TASK FORMULATION

The objective of AC power flow analysis is to determine the active power, reactive power, voltage magnitude, and voltage angle of every bus in a power network. For every bus, there is a set of known and unknown parameters, differing based on bus type (see Table 8) and the task is to estimate the unknowns. This can be modeled as a node regression task, using buses as nodes and lines as edges. Given a set of buses in a network, $\mathcal{N}$, and lines connecting them, $\mathcal{E}$, Equations 1–4 must be satisfied.

$$p_{\text{net},i} = |v_i| \sum_{j=1}^{n} |v_j|(G_{ij} \cos \theta_{ij} + B_{ij} \sin \theta_{ij}), \qquad i \in \mathcal{N}. \tag{1}$$

$$q_{\text{net},i} = |v_i| \sum_{j=1}^{n} |v_j|(G_{ij} \sin \theta_{ij} - B_{ij} \cos \theta_{ij}), \qquad i \in \mathcal{N}. \tag{2}$$

$$p_{ij} = |v_i||v_j|(G_{ij} \cos \theta_{ij} + \sin \theta_{ij}) - G_{ij}|v_i|^2, \qquad (i,j) \in \mathcal{E}. \tag{3}$$

$$q_{ij} = |v_i||v_j|(G_{ij} \sin \theta_{ij} - B_{ij} \cos \theta_{ij}) + |v_i|^2(B_{ij} - b_{s,ij}), \qquad (i,j) \in \mathcal{E} \tag{4}$$

Where:

---

[2]See https://pandapower.readthedocs.io/en/v2.0.0/powerflow/ac.html

- $p_{\text{net},i}$ is the net active power injections at bus $i$

- $q_{\text{net},i}$ is the net reactive power injections at bus $i$

- $p_{ij}$ is the active power flows on line $(i,j)$

- $q_{ij}$ is the reactive power flows on line $(i,j)$

- $|v_i|$ is voltage magnitude at bus $i$

- $\theta_i$ is the voltage phase angle at bus $i$

- $\theta_{ij}$ is the voltage angle difference between buses $i$ and $j$

- $G_{ij}$ is the real part of the admittance between buses $i$ and $j$

- $B_{ij}$ is the imaginary part of the admittance between buses $i$ and $j$

- $b_{s,ij}$ is the line shunt susceptance between buses $i$ and $j$

## D ANALYTICAL MODELS

### D.1 DC POWER FLOW FORMULATION

As an analytical baseline, we employ the DC Power Flow (DCPF) model, a widely used linearization of the AC power flow equations typically applied in transmission system analysis. We utilize the implementation provided by the *pandapower* library (Thurner et al., 2018).

The DCPF model is derived from the full AC power flow equations (Equations 1–4) by applying three simplifying assumptions:

1. **Lossless Lines:** The line resistance is negligible ($r_{ij} \approx 0$), implying $G_{ij} \approx 0$.

2. **Flat Voltage Profile:** Voltage magnitudes are fixed at $|v_i| \approx 1.0$ p.u. for all buses.

3. **Small Voltage Angles:** The phase angle differences are small, such that $\sin(\theta_i - \theta_j) \approx \theta_i - \theta_j$ and $\cos(\theta_i - \theta_j) \approx 1$.

Under these assumptions, the reactive power flows ($Q$) and voltage magnitude variations are decoupled and ignored. The system reduces to a set of linear equations relating active power injection $P_i$ to voltage angles $\theta$:

$$p_{\text{net},i} = \sum_{j \in \mathcal{N}_i} B_{ij}(\theta_i - \theta_j) = \sum_{j \in \mathcal{N}_i} \frac{\theta_i - \theta_j}{x_{ij}} \tag{5}$$

### D.2 LINDISTFLOW FORMULATION

To benchmark the performance of the proposed neural network models, we implement the Linearized Distribution Flow (LinDistFlow) approximation as an analytical baseline. This formulation is derived from the standard DistFlow equations originally introduced by Baran & Wu (1989) for radial distribution networks. LinDistFlow linearizes the non-linear AC power flow equations by neglecting second-order loss terms, providing a convex approximation suitable for voltage estimation in distribution grids.

#### D.2.1 VOLTAGE MAGNITUDE ESTIMATION

Consider a radial distribution network represented by a directed graph $\mathcal{G} = (\mathcal{N}, \mathcal{E})$, where $\mathcal{N}$ is the set of buses and $\mathcal{E}$ is the set of lines. For each line $(i,j) \in \mathcal{E}$ connecting bus $i$ to bus $j$, let $z_{ij} = r_{ij} + \mathbf{j}x_{ij}$ denote the complex impedance. Let $V_i$ be the complex voltage magnitude at bus $i$, and let $S_{ij} = P_{ij} + \mathbf{j}Q_{ij}$ be the complex power flowing from bus $i$ to bus $j$.

The original non-linear DistFlow equations describing the voltage drop and power balance are:

$$V_j^2 = V_i^2 - 2(r_{ij}P_{ij} + x_{ij}Q_{ij}) + (r_{ij}^2 + x_{ij}^2)\frac{P_{ij}^2 + Q_{ij}^2}{V_i^2} \tag{6}$$

$$P_{ij} = P_j^L + \sum_{k:(j,k)\in\mathcal{E}} \left( P_{jk} + r_{jk}\frac{P_{jk}^2 + Q_{jk}^2}{V_j^2} \right) \tag{7}$$

$$Q_{ij} = Q_j^L + \sum_{k:(j,k)\in\mathcal{E}} \left( Q_{jk} + x_{jk}\frac{P_{jk}^2 + Q_{jk}^2}{V_j^2} \right) \tag{8}$$

where $P_j^L$ and $Q_j^L$ represent the active and reactive load consumption at node $j$.

The LinDistFlow model introduces the assumption of negligible line losses. Particularly, since branch flows in p.u. are typically small, their squares are even smaller compared to the linear voltage drop components. Multiplying this by impedance squared ($r^2 + x^2$), which is also small, makes the value negligible. As a result, the quadratic terms containing ($P^2 + Q^2$) in Equations 6- 8 are ignored. Applying these simplifications yields the linear update rule for the squared voltage magnitudes:

$$V_j^2 \approx V_i^2 - 2(r_{ij}P_{ij} + x_{ij}Q_{ij}) \tag{9}$$

**Implementation:** We calculate the baseline voltages using a two-step iterative traversal of the radial tree, equivalent to a non-iterative Forward-Backward Pass:

1. **Backward Pass (Power Accumulation):** Traversing from the leaf nodes up to the root (slack bus), we aggregate the loads to determine the branch flows. For a node $j$ fed by node $i$:
$$P_{ij} = P_j^L + \sum_{k:(j,k)\in\mathcal{E}} P_{jk}, \quad Q_{ij} = Q_j^L + \sum_{k:(j,k)\in\mathcal{E}} Q_{jk} \tag{10}$$

2. **Forward Pass (Voltage Calculation):** Starting from the slack bus (where $V_{slack}$ is fixed), we traverse downstream to the leaves. We update the voltage of each child node $j$ using the voltage of its parent $i$ and the branch flows calculated in the previous step, via Equation 9.

### D.2.2 VOLTAGE ANGLE RECOVERY

While the original Baran and Wu formulation focused on voltage magnitudes and losses, the voltage angles can be recovered using a complementary linearization often utilized in convex relaxations of Optimal Power Flow, as detailed by Farivar & Low (2013), and a similar set of common assumptions.

Starting from the voltage drop equation $V_j = V_i - (r_{ij} + \mathbf{j}x_{ij})I_{ij}$ and applying the small-angle approximation ($\sin(\theta_{ij}) \approx \theta_i - \theta_j$ and $\cos(\theta_{ij}) \approx 1$), the voltage angle difference is decoupled from the voltage magnitude. The resulting linear relationship is:

$$\theta_j \approx \theta_i - \frac{x_{ij}P_{ij} - r_{ij}Q_{ij}}{V_{nom}^2} \tag{11}$$

where $\theta$ is the voltage angle in radians and $V_{nom}$ is the nominal voltage (typically approximated as the slack bus voltage magnitude).

**Implementation:** Using the same branch flows $P_{ij}$ and $Q_{ij}$ computed during the Backward Pass, we perform a Forward Pass to update the angles. Note that unlike the magnitude update which depends on $rP + xQ$, the angle update is driven by the cross-coupling term $xP - rQ$.

### D.2.3 PREDICTIVE PERFORMANCE

We present the LinDistFlow-derived voltage prediction results alongside those of our best performing models and the DC Power Flow analytical baseline. Figure 4 illustrates that utilizing this analytical baseline for voltage magnitude and angle prediction is not competitive when compared to our physics-informed GNN solvers. One explanation for the significant errors could be due to the fact that heavily loaded lines violate the "negligible loss" assumption of standard LinDistFlow, as they

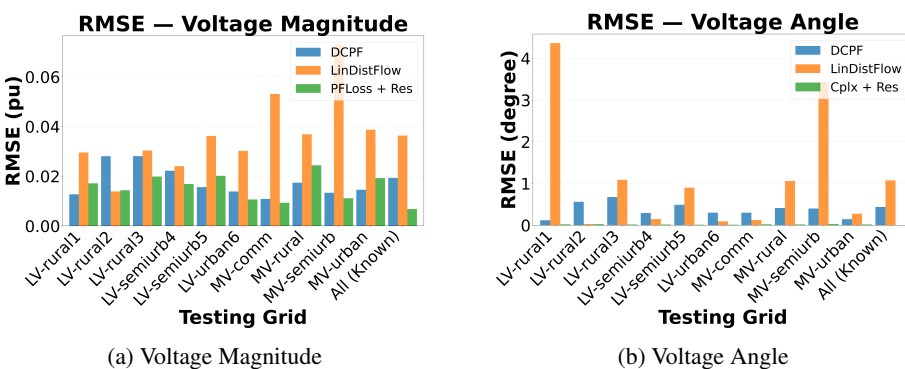

(a) Voltage Magnitude            (b) Voltage Angle

Figure 4: Comparison of Best-Performing models to DC PF and LinDistFlow.

cause the quadratic loss term to be a more significant driver of voltage drops. As seen in Table 6, some lines in the dataset networks have higher line loadings which would lead to large errors on the most stressed lines, which would then propagate this miscalculated voltage down the network. The superior performance of the flat-voltage baseline (DCPF) over the analytical LinDistFlow baseline highlights the difficulty of applying simplified physical models to resistive, low-voltage grids with distributed generation. Here, the model that predicts "no change" (i.e., a flat voltage profile) beats a physics-based model that relies on imperfect linearizations. This motivates the need for data-driven approaches such as our physics-informed GNN solvers that can learn the non-linear relationship between injection and voltage without relying on ideal-grid assumptions.

Table 9: Raw Hybrid Model Performance Results by Testing Grid

| Testing Grid | Model | RMSE VM | RMSE VA | Train (s) | Inference (ms) | Capacity |
|---|---|---|---|---|---|---|
| LV-rural1 | Cplx + PFLoss | 0.05899 | 0.03190 | 5326.4 | 6.66906 | 603.4K |
| LV-rural1 | Cplx + Res | 0.04141 | **0.02177** | 5114.9 | 6.70865 | 603.4K |
| LV-rural1 | PFLoss + Res | **0.01713** | 2.59823 | **1296.0** | **1.97338** | 302.7K |
| LV-rural2 | Cplx + PFLoss | 0.02692 | 0.02712 | 4993.6 | 6.71781 | 603.4K |
| LV-rural2 | Cplx + Res | **0.01266** | **0.02517** | 4711.0 | 6.74455 | 603.4K |
| LV-rural2 | PFLoss + Res | 0.01431 | 1.53923 | **1321.8** | **1.98800** | 302.7K |
| LV-rural3 | Cplx + PFLoss | 0.01572 | 0.01615 | 4655.2 | 6.60150 | 603.4K |
| LV-rural3 | Cplx + Res | **0.01567** | **0.01559** | 4562.5 | 6.61441 | 603.4K |
| LV-rural3 | PFLoss + Res | 0.01984 | 0.95270 | **1398.8** | **1.95619** | 302.7K |
| LV-semiurb4 | Cplx + PFLoss | 0.01762 | **0.01314** | 5040.8 | 6.86442 | 603.4K |
| LV-semiurb4 | Cplx + Res | **0.01642** | 0.01643 | 4952.4 | 6.84558 | 603.4K |
| LV-semiurb4 | PFLoss + Res | 0.01685 | 0.74623 | **1269.5** | **2.02061** | 302.7K |
| LV-semiurb5 | Cplx + PFLoss | 0.02258 | 0.01119 | 4751.3 | 6.69333 | 603.4K |
| LV-semiurb5 | Cplx + Res | 0.02161 | **0.01033** | 4603.7 | 6.93617 | 603.4K |
| LV-semiurb5 | PFLoss + Res | **0.02015** | 0.63628 | **1319.5** | **2.08807** | 302.7K |
| LV-urban6 | Cplx + PFLoss | 0.00952 | **0.01069** | 4990.5 | 7.24550 | 603.4K |
| LV-urban6 | Cplx + Res | **0.00923** | 0.01137 | 4885.8 | 7.17983 | 603.4K |
| LV-urban6 | PFLoss + Res | 0.01061 | 0.73603 | **1280.6** | **2.12642** | 302.7K |
| MV-comm | Cplx + PFLoss | 0.14504 | 0.29959 | 4735.3 | 6.94522 | 603.4K |
| MV-comm | Cplx + Res | **0.00828** | **0.02088** | 4632.4 | 6.76833 | 603.4K |
| MV-comm | PFLoss + Res | 0.00929 | 0.90102 | **1339.4** | **1.98995** | 302.7K |
| MV-rural | Cplx + PFLoss | **0.01440** | **0.00984** | 4776.8 | 6.74322 | 603.4K |
| MV-rural | Cplx + Res | 0.03721 | 0.01876 | 4687.5 | 6.73881 | 603.4K |
| MV-rural | PFLoss + Res | 0.02438 | 0.67677 | **1273.8** | **1.99153** | 302.7K |
| MV-semiurb | Cplx + PFLoss | 0.01444 | 0.03327 | 4669.2 | 6.67193 | 603.4K |
| MV-semiurb | Cplx + Res | 0.01941 | **0.02981** | 4583.0 | 6.69604 | 603.4K |
| MV-semiurb | PFLoss + Res | **0.01108** | 1.62419 | **1286.6** | **1.97380** | 302.7K |
| MV-urban | Cplx + PFLoss | **0.01680** | 0.01259 | 4598.4 | 6.56538 | 603.4K |
| MV-urban | Cplx + Res | 0.01866 | **0.01821** | 4484.1 | 6.54877 | 603.4K |
| MV-urban | PFLoss + Res | 0.01923 | 1.10563 | **1335.9** | **1.95899** | 302.7K |
| All (Known) | Cplx + PFLoss | 0.00585 | **0.00251** | 5147.2 | 6.76416 | 603.4K |
| All (Known) | Cplx + Res | **0.00580** | 0.00254 | 4759.2 | 6.76174 | 603.4K |
| All (Known) | PFLoss + Res | 0.00677 | 0.16124 | **1554.3** | **2.00352** | 302.7K |

Table 10: Raw GAT-based Model Performance Results by Testing Grid

| Testing Grid | Model | RMSE VM | RMSE VA | Train (s) | Inference (ms) | Capacity |
|---|---|---|---|---|---|---|
| LV-rural1 | GAT | **0.02692** | 0.72323 | 2023.3 | 4.23116 | 304.5K |
| LV-rural1 | GAT-Cplx | 22.42461 | 0.60843 | 6391.2 | 9.99332 | 605.2K |
| LV-rural1 | GAT-PFLoss | 0.07893 | **0.49025** | 2122.9 | **4.21070** | 304.5K |
| LV-rural1 | GAT-Res | 0.05316 | 8.25658 | **1847.7** | 4.23108 | 304.5K |
| LV-rural2 | GAT | 0.16296 | 2.84787 | 1986.7 | 4.30918 | 304.5K |
| LV-rural2 | GAT-Cplx | **0.02327** | **0.02832** | 5911.2 | 10.28349 | 605.2K |
| LV-rural2 | GAT-PFLoss | 0.14166 | 2.61067 | 2082.3 | **4.29760** | 304.5K |
| LV-rural2 | GAT-Res | 0.06558 | 4.88404 | **1914.1** | 4.32212 | 304.5K |
| LV-rural3 | GAT | 0.01785 | 0.43163 | 2503.7 | 4.25854 | 304.5K |
| LV-rural3 | GAT-Cplx | 0.01781 | **0.01423** | 5729.4 | 10.19612 | 605.2K |
| LV-rural3 | GAT-PFLoss | 0.03165 | 0.63153 | 2602.0 | **4.25799** | 304.5K |
| LV-rural3 | GAT-Res | **0.01658** | 0.70123 | **1892.8** | 4.28400 | 304.5K |
| LV-semiurb4 | GAT | 0.01657 | 0.86837 | 1925.5 | 4.33177 | 304.5K |
| LV-semiurb4 | GAT-Cplx | 0.02481 | **0.01070** | 5128.1 | 10.36799 | 605.2K |
| LV-semiurb4 | GAT-PFLoss | 0.01557 | 0.95384 | 2024.9 | **4.30347** | 304.5K |
| LV-semiurb4 | GAT-Res | **0.01433** | 1.23636 | **1833.2** | 4.34664 | 304.5K |
| LV-semiurb5 | GAT | 0.02038 | 0.55621 | 2133.0 | 4.50983 | 304.5K |
| LV-semiurb5 | GAT-Cplx | 0.01794 | **0.00968** | 5829.4 | 10.74776 | 605.2K |
| LV-semiurb5 | GAT-PFLoss | **0.01440** | 0.47729 | 2231.9 | **4.32165** | 304.5K |
| LV-semiurb5 | GAT-Res | 0.01574 | 0.77447 | **2057.1** | 4.64251 | 304.5K |
| LV-urban6 | GAT | 0.01228 | 0.70808 | 1786.1 | **4.31373** | 304.5K |
| LV-urban6 | GAT-Cplx | 0.01043 | **0.01152** | 6139.9 | 10.78284 | 605.2K |
| LV-urban6 | GAT-PFLoss | 0.01277 | 0.67939 | 1885.3 | 4.45540 | 304.5K |
| LV-urban6 | GAT-Res | **0.01003** | 0.70414 | **1781.5** | 4.54937 | 304.5K |
| MV-comm | GAT | 0.01703 | 1.94529 | 2108.3 | 4.29743 | 304.5K |
| MV-comm | GAT-Cplx | 0.03439 | **0.04264** | 5854.8 | 10.23937 | 605.2K |
| MV-comm | GAT-PFLoss | **0.00921** | 1.86529 | 2205.3 | **4.27714** | 304.5K |
| MV-comm | GAT-Res | 0.01192 | 2.75916 | **1473.5** | 4.31093 | 304.5K |
| MV-rural | GAT | 0.02162 | 0.72117 | 2358.1 | 4.30109 | 304.5K |
| MV-rural | GAT-Cplx | **0.01797** | **0.01438** | 5903.9 | 10.32823 | 605.2K |
| MV-rural | GAT-PFLoss | 0.01910 | 0.68771 | 2454.5 | **4.29609** | 304.5K |
| MV-rural | GAT-Res | 0.02196 | 1.16897 | **1314.6** | 4.31832 | 304.5K |
| MV-semiurb | GAT | 0.03349 | 3.59725 | 1944.9 | 4.29249 | 304.5K |
| MV-semiurb | GAT-Cplx | 0.01091 | **0.02834** | 5747.4 | 10.40490 | 605.2K |
| MV-semiurb | GAT-PFLoss | 0.02644 | 3.23623 | 2040.5 | **4.28757** | 304.5K |
| MV-semiurb | GAT-Res | **0.00984** | 1.68319 | **1548.8** | 4.30757 | 304.5K |
| MV-urban | GAT | 0.01215 | 1.05314 | **1896.6** | **4.24654** | 304.5K |
| MV-urban | GAT-Cplx | **0.01087** | **0.01309** | 5664.7 | 10.70423 | 605.2K |
| MV-urban | GAT-PFLoss | 0.01851 | 1.01688 | 1994.8 | 4.31551 | 304.5K |
| MV-urban | GAT-Res | 0.02866 | 1.53854 | 2114.5 | 4.49710 | 304.5K |
| All (Known) | GAT | 0.00981 | 0.15869 | **2062.0** | 4.30639 | 304.5K |
| All (Known) | GAT-Cplx | **0.00484** | **0.00236** | 5950.3 | 10.21571 | 605.2K |
| All (Known) | GAT-PFLoss | 0.00822 | 0.15646 | 2162.0 | **4.29806** | 304.5K |
| All (Known) | GAT-Res | 0.01402 | 0.16708 | 2188.4 | 4.31734 | 304.5K |