# OpenReview forum: "Physics-Informed Inductive Biases for Voltage Prediction in Distribution Grids"
_ICLR.cc/2026/Conference — Submitted to ICLR 2026_

### Official Review · Reviewer_iiq2 · 2025-10-31

**Soundness:** 2
**Presentation:** 3
**Contribution:** 1
**Rating:** 2
**Confidence:** 5

**Summary:**

The paper studies three ways to include “physics” into GNN voltage predictors for distribution grids: (i) fine-tuning with a physics loss (power balance equation), (ii) using complex-valued GNNs to represent voltages naturally, and (iii) predicting residuals relative to a baseline solver. The authors evaluate these on a test suite derived from SimBench and report that physics-loss helps magnitude prediction, where complex nets give higher angle accuracy, and residuals are mixed. They claim these are “inductive biases” that improve OOD generalization.

**Strengths:**

The papers focus is on OOD problem, which is very relevant for the distribution grids as topological changes occur frequently due to line switchings in distribution grids. Plus, ENGAGE dataset is claimed to be having multi-grid testbed rather than the single feeder which represents practical conditions.  Writeup is also reasonably clear.

**Weaknesses:**

Major Weaknesses:
1. **No conceptual novelty:** all three ways to put physics into the problem are already known as cited by authors. Presenting them together is more of an engineering comparison, not a new inductive bias theory or a fundamentally new method. The paper’s framing as discovering a new “inductive bias” is kind of confusing and looks a bit overstating the claim; they are conflate architecture choice (a true inductive bias reduction thing) with training regularizers and output reparameterizations (which are optimization/design choices).
2. **Methodological Error:** DC power flow assumes voltage magnitudes are fixed (typically 1.0 p.u.) and linearizes only active power as a function of voltage angles (See DC Power Flow Revisied by B. Stott et.al.). As it does not model or predict voltage magnitudes, using DCPF as a baseline for magnitude RMSE is therefore meaningless (it either trivially treats |V|=1 and measures deviation from 1.0, or it applies an ad-hoc, unsupported hack to derive magnitudes). There is a very very rich theory of power flow approximations which authors overlooked completely (Dan Molzahn et.al A Survey of Relaxations and Approximations of the Power Flow Equations). Authors must justify or remove that comparisons as they are not valid.

The statement : "a commonly used solving method for voltage prediction: DC Power Flow" is incorrect.

**2(a)**. Further, as per the description of ENGAGE in paper, DCPF cannot be applied even to predict the voltage angles on authors test cases.  For low-voltage and many distribution feeders the R/X ratio is high, so the common DC assumptions (negligible R, small angle differences, and that reactive flows can be ignored) break down. Using DC approximations as a performance yardstick for distribution/low-voltage problems is therefore inappropriate unless the authors explicitly justify or quantify the DC error in those regimes. Distribution literature offers linear models suited to voltage magnitude estimation (DistFlow / LinDistFlow), which are a more appropriate classical baseline. The authors reference that linear methods are unreliable in distribution grids, but do not quantify or show the DC vs AC discrepancy across the ENGAGE grids.

3. **Missing Baselines:**  There is a very large volume of work on power flow predictions. The paper compares against a “baseline GNN” and its variants, but omits credible DNN baselines and Non-DNN Gaussian Process regression (and sparse GPs),  and very importantly linearized analytic surrogates (LinDistFlow, parameterized linear models).  GPs in particular are an established surrogate for voltage mapping and probabilistic power flow; they often perform strongly in low-data regimes and give calibrated uncertainties — similar to the setting the authors claim to improve. (See B. Tan et. al. Gaussian processes in power systems: Techniques, applications, and future works).

**Questions:**

1. How does complex-valued DNN scale? What are the sizes of networks in the ENGAGE Dataset along with other electrical properties like R/X ratio?
2. Can formal justification be provided that how all three components presented reduce the hypothesis space or inductive bias?
3. How do you distinguish between improved optimization (better loss landscape) and genuine inductive bias (systematic generalization) in your results?
4. Why do you fine-tune for only 20 epochs with the physics-informed loss? Was this empirically optimal? How about using Primal Dual kind of methods? (I know they are generally designed for OPF but the constraint of power balance is same).
5. Can you test on PGLib cases? As your method doesn't take any network specific assumption, will it work for large scale mesh grids?
6. How exactly did you compute voltage-magnitude RMSE for the DC power-flow baseline? Is DCPF formulation applicable to your LV grid test cases?
7. Some reported errors are <10⁻³ p.u. or <0.002°. Are such differences meaningful given sensor precision and modeling uncertainty? What error is a good error for power flow problem?

---

> ### Author Response · Authors · 2025-11-21
> **Response to Reviewer iiq2**
>
> We thank the reviewer for their rigorous technical feedback and particularly for the application-driven assessment of our paper. We have parsed your detailed review and made sure to address all of it. Below are responses to the concerns regarding novelty, relevant baselines such as LinDistFlow, and provided clarifications around specific details of our study. Additionally, **we provide a PDF for figure and table references** (_only_ additional experiments and new manuscript modifications): https://figshare.com/s/78e08be73a5636987e78.
>
> ### Novelty and "Discovery"
>
> We acknowledge that the paper’s original framing was ambiguous. We strictly do not claim to have discovered new biases. As stated in our introduction, our goal is a "systematic assessment of physics-informed inductive biases." We have revised the manuscript to ensure this distinction is absolutely clear. Previous works [1, 2] motivate and claim the generalization potential of physics informed machine learning and this technique is commonplace in current ML applications to power systems, but there has been neither a consistent benchmark nor a dedicated generalization assessment here. Therefore,  **our core contribution is to address the lack of comparability in the field through the rigorous validation and benchmarking of voltage prediction inductive biases**. Furthermore, we contribute the first CVNN architecture for distribution grid voltage prediction and demonstrate its generalization advantage, providing a new insight. This has now been made clearer in our manuscript. Finally, guided by the highly useful feedback, we add initial hybrid inductive bias results (from future works) to the final manuscript (see Section B.1 of the linked PDF).
>
> We hope the reviewer agrees how valuable comparable benchmarks are in enhancing the maturity of machine learning in power systems and can thus acknowledge our structured attempt to validate a common category of approaches that claim to improve model robustness.
>
> ### DCPF and Baselines
>
> We agree with the reviewer's expert assessment that DCPF is a highly simplified and physically inaccurate model for distribution grids. We have corrected the confusing formulation “commonly used solving method” to “commonly used simplification of the full AC Power Flow solution” and clarified the |V|=1 assumption. DCPF was not used as a physically meaningful baseline, but rather as an **empirically observed baseline for generalization failure**, as established in prior benchmarking work (e.g. [3, 4]). We use it to quantify the generalization performance barrier: if a sophisticated physics-informed GNN cannot perform better than an approximation that ignores all reactive power and fixes voltage magnitude to 1.0 p.u., then the GNN is not fit for OOD application. Therefore, its inclusion is a deliberate check on model stability and fitness, not a methodological error or core contribution.
>
> However, as suggested, **we will benchmark LinDistFlow as an important analytic baseline** to compare generalization performance. As LinDistFlow is a more sophisticated model than DC PF, we believe it will quantify how large the generalization gap truly is of these machine learning models, thus further emphasizing the importance of dedicated generalization studies. This experimentation is ongoing and will be added to the final manuscript.
>
> ### Optimization vs Inductive Bias
> We believe all represent genuine inductive bias rather than simply improved optimization.
> - Loss: We explicitly decouple optimization from bias using a two-stage training process. The model is first fully trained using the standard MSE loss (optimization). The subsequent, short physics-informed fine-tuning is not a re-optimization but a targeted structural refinement of the already-optimized weights. We are effectively projecting the learned function onto the manifold of physically consistent solutions, rather than just facilitating a better optimization path.
> - Complex: The inductive bias is inherent in the architectural representation itself, independent of the loss function or optimization path. By representing voltages and impedances as complex numbers, the architecture intrinsically couples the magnitude and phase information. This enforces the fundamental constraints of a single, dependent voltage output, thereby restricting the hypothesis space to only include functions that operate on coupled electrical quantities.
> - Residuals: Instead of forcing the GNN to learn the absolute, highly non-linear voltage profile V, we try to learn only the residual correction relative to a known, simplified solution (e.g. a simple baseline solver or fixed 1.0 p.u. voltage). This reframing significantly reduces the complexity of the target function the neural network must learn, biasing the solution toward solutions near the known coarse approximation. This simplification of the target function space constitutes a genuine inductive bias, and it has been claimed to help learning [5].

---

> > ### Author Response · Authors · 2025-11-21
> > **Response to Reviewer iiq2 (continued)**
> >
> > ### Fine-Tuning and Primal-Dual Methods
> > - The 20-epoch limit was deliberately chosen to ensure **this phase acts solely as a structural refinement**, preventing it from becoming a deep re-optimization of the landscape. Since crucial structural adjustments require minimal steps on a pre-trained model, we selected this value because it represents <1% of the total training epochs. We observed no significant additional gain in OOD performance when extending this phase up to ex. 30 epochs.
> > - We agree that Primal-Dual methods are ideal for constrained optimization (OPF) but the physics-informed loss regularization is the appropriate method for our supervised prediction task, consistent with existing power flow ML literature. **Since we aim to validate existing generalization claims, we opt for the more standard method.**
> >
> > ### ENGAGE vs PGLib
> > PGLib and standard IEEE transmission cases are great benchmarks for transmission systems (meshed, high voltage, low R/X). However, **our work is explicitly focused on distribution systems which face distinct challenges**. Therefore, the ENGAGE dataset is the appropriate benchmark for this domain. Within the ENGAGE dataset there are 10 distinct distribution networks from size 15 to 136. **We detail the ENGAGE dataset in Sections B.3 and B.4 of the linked PDF** and clarify this in the final manuscript.
> >
> > ### Real-World Interpretation
> > This gets to heart of the motivation for generalization studies: to improve the applicability of research results for real-world distribution grid operation. As such, **we improvde the discussion to provide practical understanding of the quality of the model performances (Section B.5 of the linked PDF)**. We show that in-distribution performance of all models fall within the practical noise floor, making them acceptable for operational in known grids (Table 2 of our original manuscript), however most are unfit for practical application when confronted with unseen grids. Our results illustrate that there should be a shift in machine learning modeling focus from standard predictive accuracy to generalization performance to make meaningful steps towards real-world application.
> >
> >
> > [1] George Em Karniadakis, Ioannis G Kevrekidis, Lu Lu, Paris Perdikaris, Sifan Wang, and Liu Yang. Physics-informed machine learning. Nature Reviews Physics, 3(6):422–440, 2021.
> > [2] Bin Huang and Jianhui Wang. Applications of Physics-Informed Neural Networks in Power Systems - A Review. IEEE Transactions on Power Systems, 38(1):572–588, January 2023. ISSN 1558-0679. doi: 10.1109/TPWRS.2022.3162473.
> > [3] Ehimare Okoyomon and Christoph Goebel. A framework for assessing the generalizability of gnnbased ac power flow models. In Proceedings of the 16th ACM International Conference on Future and Sustainable Energy Systems, pp. 476–488, 2025.
> > [4] Jonas Berg Hansen, Stian Normann Anfinsen, and Filippo Maria Bianchi. Power Flow Balancing With Decentralized Graph Neural Networks. IEEE Transactions on Power Systems, 38(3): 2423–2433, May 2023. ISSN 1558-0679. doi: 10.1109/TPWRS.2022.3195301.
> > [5] Za'ter, Muhy Eddin, Bri-Mathias Hodge, and Kyri Baker. Residual Correction Models for AC Optimal Power Flow Using DC Optimal Power Flow Solutions. arXiv preprint arXiv:2510.16064 (2025).

---

> > ### Author Response · Authors · 2025-12-03
> > **DCPF vs LinDistFlow**
> >
> > We have implemented and benchmarked LinDistFlow, as a response to the reviewer's request for a more appropriate baseline.
> >
> > **Our results show that the LinDistFlow baseline is not competitive for voltage magnitude and angle prediction in our test suite when compared to our physics-informed GNN solvers and DC PF**.
> >
> > One explanation could be because heavily loaded lines violate the "negligible loss" assumption of standard LinDistFlow, as they cause the quadratic loss term to be a more significant driver of voltage drops. Here, the model that predicts "no change" (i.e., DC PF with its flat voltage profile) beats a physics-based model that relies on imperfect linearizations. This motivates the need for data-driven approaches such as our physics-informed GNN solvers that can learn the non-linear relationship between injection and voltage without relying on ideal-grid assumptions.

---

### Official Review · Reviewer_M86w · 2025-10-31

**Soundness:** 2
**Presentation:** 3
**Contribution:** 2
**Rating:** 2
**Confidence:** 4

**Summary:**

This paper investigates how different physics-informed inductive biases influence the performance and generalization of graph-based models for voltage prediction in electrical distribution grids.  The authors focus on three types of inductive biases, Power-flow-constrained loss functions, Complex-valued neural networks, Residual-based task reformulation. The study measures both in-distribution and out-of-distribution (OOD) generalization via leave-one-grid-out experiments. Results show that the physics-informed loss variant achieves the most stable voltage-magnitude prediction and lowest variability across unseen topologies.

**Strengths:**

- The paper provides a well-controlled setup: all models share identical GNN architectures, training schedules, and datasets, allowing fair isolation of each inductive bias’s effect.
- The paper is well organized and provides concrete implementation details, including formulas for the power-flow loss, CVNN layer definitions, and hyperparameters.

**Weaknesses:**

- The study mainly benchmarks three established strategies. None introduces a new algorithmic formulation, architecture, or theoretical insight. The contribution is primarily empirical rather than methodological.
- The study evaluates all inductive biases on a single GraphConv backbone. However, power-flow learning performance can vary substantially across graph architectures (e.g., GCN, GAT, Graph Transformers). Without cross-architecture validation, it is difficult to attribute improvements to the inductive bias itself rather than to properties of GraphConv.
- Experimental results should perform a more comprehensive comparison, like model capacity, convergence speed, and inference efficiency. Compared to different archs, even traditional methods.
- While the paper claims to study “physics-informed inductive biases,” the bias is imposed only at the loss level or representation level, not within the GNN’s message-passing structure. There is no attempt to embed Kirchhoff’s laws, admittance matrices, or nodal couplings directly into the propagation mechanism.
- The ENGAGE dataset is derived solely from SimBench; it is unclear how many unique topologies exist or how dissimilar the held-out grids truly are. A stronger test would involve training on small IEEE systems (14-bus, 39-bus) and evaluating on larger ones (57-, 118-bus)** to demonstrate scale generalization.
- The paper would benefit greatly from schematic diagrams showing the overall experimental setup, the flow of each inductive bias, or a qualitative visualization of predicted vs. true voltage profiles.
- Large sections of the paper read as tutorial material. Yet the study lacks a strong discussion on how the proposed findings translate to real distribution-grid operations

**Questions:**

Personally, the paper primarily benchmarks three established inductive-bias strategies, Please authors clarify what the core methodological contribution or new insight is beyond this comparative study.

---

> ### Author Response · Authors · 2025-11-20
> **Response to Reviewer M86w**
>
> We thank the reviewer for their detailed assessment. We address the concerns below, with a particular focus on novelty, scale generalization, and architecture validation. Additionally, **we provide a PDF for figure and table references** (_only_ additional experiments and new manuscript modifications): https://figshare.com/s/78e08be73a5636987e78.
>
> ### Novelty and Contribution
> While we value new methodologies for reliable grid operation, many studies introduce models with curated tests and non-public benchmarks (e.g., [1,2]), which **has led to a major challenge in comparability**. Our work fills this critical gap by objectively assessing **which design choices truly confer systematic generalization**, providing a necessary contribution for the maturity of the field. Thus, we emphasize that our paper’s core contribution is not the introduction of new biases, but rather the rigorous **systematic validation and benchmarking** of existing, unverified claims regarding the benefit of “physics-informed machine learning” for generalization [3, 4]. By evaluating physics-informed inductive biases on a neutral multi-grid benchmark with a consistent GNN backbone, we aim to establish a reliable basis for comparison, and thus take an active step towards creating a **consistent performance baseline** for generalization, allowing researchers to confidently assess new methodologies against established, validated design choices.
>
> As indicated in our related works, complex-valued neural networks are not well-represented in current power systems literature. We contribute the first CVNN architecture for distribution grid voltage prediction and demonstrate its generalization advantage, providing a new insight. This has now been made clearer in our manuscript. Also, guided by this systematic analysis, we investigate several hybrid inductive bias models which will be added to our final manuscript (see **Section B.1 of the linked PDF**).
>
> Ultimately, we establish the first comprehensive evaluation of OOD generalization in GNN-based power-flow prediction, demonstrating that **reliability for real distribution system deployments cannot be judged from in-distribution accuracy alone but must be grounded in systematic generalization analysis**.
>
> ### Scale Generalization & OOD Setup
> 1. The ENGAGE dataset is heterogeneous, with 10 unique base distribution grids with varying sizes (15 to 136). Thus, **our "leave-one-out" protocol already tests scale generalization**. The suggested IEEE  systems, though established benchmarks for transmission systems, have fundamentally different characteristics than distribution networks which are the focus of our study. We will clarify this in the final manuscript and detail specific characteristics about the ENGAGE dataset (see Sections B.3 and B.4 of the linked PDF).
> 2. Though a one-to-one grid evaluation is possible, previous studies [1, 2, 5, 6] indicate that SOTA ML methods are immature for this one-to-one generalization case. As such, we focus on the OOD setting for our generalization study. **We have now clarified this as well in the final manuscript.**
>
> ### Cross-Architecture Validation
> We acknowledge the reviewer’s concern for a more robust separation between GNN architecture and inductive bias. **We have re-run the baseline and all single inductive-bias models on a GAT backbone**. Section B.2 of the linked PDF confirms the efficacy of these inductive biases. It is shown that we can observe non-architecturally specific trends, validating that the generalization benefits stem from the biases themselves, not the specific GNN backbone.
>
> ### Scope of Physics Integration
>
> Physics-informed message passing is a very interesting direction. However, we present three common inductive biases to verify existing claims. As such, we intentionally fix the message passing structure to isolate the bias's effect.
>
> ### Metrics and Diagrams
> We have added a schematic diagram (Figure B.1 of the linked PDF) and will add extra results for model capacity and inference time in the final manuscript.
>
> ### Real-World Interpretation of Results
> This comment is particularly useful for the interpretation of our results and the main takeaways of the paper. We have improved the discussion to provide practical understanding of the quality of the model performances (see Section B.5 of the linked PDF). We show that the in-distribution performance of all models fall within the practical noise floor, making them acceptable for operational in known distribution grids (Table 2 of our original manuscript). However, the importance of such generalization studies is highlighted when evaluating unseen grids. **We observe that performance largely deteriorates, resulting in models unfit for practical application**. OOD performance thus becomes a distinguishing feature and by **using physics-informed-inductive-biases we can close this generalization gap further** and take steps towards truly robust and reliable voltage prediction models.

---

> ### Author Response · Authors · 2025-11-20
> **References**
>
> [1] Luis Bottcher, Hinrikus Wolf, Bastian Jung, Philipp Lutat, Marc Trageser, Oliver Pohl, Andreas Ulbig, and Martin Grohe. Solving AC Power Flow with Graph Neural Networks under Realistic Constraints. In 2023 IEEE Belgrade PowerTech, pp. 1–7, June 2023. doi: 10.1109/PowerTech55446.2023.10202246.
> [2] Jonas Berg Hansen, Stian Normann Anfinsen, and Filippo Maria Bianchi. Power Flow Balancing With Decentralized Graph Neural Networks. IEEE Transactions on Power Systems, 38(3): 2423–2433, May 2023. ISSN 1558-0679. doi: 10.1109/TPWRS.2022.3195301.
> [3] George Em Karniadakis, Ioannis G Kevrekidis, Lu Lu, Paris Perdikaris, Sifan Wang, and Liu Yang. Physics-informed machine learning. Nature Reviews Physics, 3(6):422–440, 2021.
> [4] Bin Huang and Jianhui Wang. Applications of Physics-Informed Neural Networks in Power Systems - A Review. IEEE Transactions on Power Systems, 38(1):572–588, January 2023. ISSN 1558-0679. doi: 10.1109/TPWRS.2022.3162473.
> [5] Ehimare Okoyomon and Christoph Goebel. A framework for assessing the generalizability of GNN-based AC power flow models. In Proceedings of the 16th ACM International Conference on Future and Sustainable Energy Systems, pp. 476–488, 2025.
> [6] Ashkan B. Jeddi and Abdollah Shafieezadeh. A Physics-Informed Graph Attention-based Approach for Power Flow Analysis. In 2021 20th IEEE International Conference on Machine Learning and Applications (ICMLA), pp. 1634–1640, December 2021. doi: 10.1109/ICMLA52953.2021. 00261.

---

### Official Review · Reviewer_M7Jo · 2025-10-31

**Soundness:** 3
**Presentation:** 3
**Contribution:** 2
**Rating:** 6
**Confidence:** 4

**Summary:**

This paper evaluates the impact of three different inductive biases for machine learning and AC power flow: physics-informed losses, complex-valued neural networks, and residual predictions. Experiments are conducted on several low- and medium-voltage distribution grids.

**Strengths:**

* The paper considers distribution grids, which have received less attention than transmission grids in the literature. The distinction is relevant, as distribution grids often display different topological structure (e.g radial vs meshed) and different behavior (e.g. 3-phased unbalanced, large deviations from nominal voltage) compared to transmission grid
* In a literature of mostly isolated works, this paper identifies three building blocks and evaluates each one's impact on training and performance.
* Of the 3 inductive biases, complex-valued neural-networks is the one I have seen the least used in the literature on ML for AC power flows

**Weaknesses:**

* The paper provides very few mathematical details (there are a total of 7 equations in the paper), which would have been especially valuable for understanding the experiment setup
* The various inductive biases are not new, however the paper does a good job of motivating and evaluating each of them
* The numerical experiments only consider one variant per inductive bias, which leaves the question of whether combining them may result in improved performance. Additional experiments that combine them would be valuable

**Questions:**

* Please provide more information on numerical experiment settings
* Please conduct additional experiments that combine two or three inductive biases
* Please provide a more comprehensive mathematical framework, including: statement of power flow equations, description of input and output features, and any information related to experiment settings

---

> ### Author Response · Authors · 2025-11-20
> **Response to Reviewer M7Jo**
>
> We thank the reviewer for their constructive feedback and for recognizing the value of our work in systematically evaluating inductive biases for (often underrepresented) distribution grids. We have addressed the request for more mathematical rigor and, crucially, have conducted the suggested experiments on combining inductive biases, which slightly improved on previous results, while introducing new hybrid models that have not yet been tested for distribution grid voltage prediction.
>
> To accompany the response, we provide an anonymous PDF for figure and table references (containing _only_ additional experiments and new manuscript modifications): https://figshare.com/s/78e08be73a5636987e78.
>
> ### 1. Combining Inductive Biases
>
> We agree that combining biases can provide a promising direction. As a matter of fact, we had conducted some prior analysis on this and had found, as indicated in the paper, that “early results suggest such benefits are non-additive and grid-specific”. That being said, the reviewer is correct that these additional experiments add value in assessing the performance of the presented inductive biases. Following this suggestion, **we implement and evaluate several hybrid models**. As detailed in **Table 1 of the linked revision PDF**, augmenting the complex and physics models with the residual-learning technique enhances the performance of each of these models, resulting in the best performance for their respective subtasks (voltage magnitude prediction for Phys-Loss + Residuals, and voltage angle prediction for Complex + Residuals). However, combining the previously-best-performing Phys-Loss and Complex models does not yield any added benefit, and results in a model that generally weakens the advantage of each original bias.
> **For more details and a draft of the additions, please see Section B.1 of the linked revision PDF.**
>
> Importantly, these hybrid models were discovered as a direct result of this benchmarking, and these findings have been added to the "Results" section as a contribution derived from this systematic study.  The resulting systematic analysis of inductive biases serves as the first comprehensive framework of its kind, demonstrating the **necessity for evaluating machine learning power flow prediction models based on their topological generalization capabilities** rather than in-distribution accuracy alone to assess whether they meet reliability standards of real distribution systems deployments.
>
> ### 2. Mathematical Framework & Details; Numerical Experiment Settings
>
> Thanks for pointing out the lack of formal descriptions of the learning task, model architectures, and datasets in the initial submission. For the final manuscript, we will expand the Methodology and Appendix to include:
> - Equations: More comprehensive mathematical formulation including the description of the Full AC Power Flow task
> - GNN Architecture: Clearer definitions of model architecture, inputs and outputs
> - Data Objects: A description of the datatype objects and their features
> - Dataset overview: Descriptors of the ENGAGE dataset, include network sizes, voltage level, type (rural, semi-urban, urban, commercial), average R/X ratio, etc.
>
> **Please see Sections B.3 and B.4 of the linked revision PDF for a draft of the additions.**

---

### Author Response · Authors · 2025-12-03
**Contribution Summary & Manuscript Revisions**

**Contribution Summary**

We thank the three reviewers once again for their detailed and highly valuable feedback on our paper. While the review from Reviewer M7Jo was largely positive, the critical feedback from Reviewers M86w and iiq2 highlighted some ambiguity in the presentation and scope of our primary contributions. This allowed us to reflect on our original paper's presentation, prompting further improvements and clarifications to our study. The results of these refinements are present in our revised manuscript and were detailed in our reviewer responses. For consistency, we reiterate our contributions here:

- We provide a systematic assessment of standard accuracy and out-of-distribution generalization of physics-informed inductive biases for voltage prediction across heterogeneous distribution networks.
- We contribute the first complex-valued neural network architecture for distribution grid voltage prediction and demonstrate its generalization advantage.
- We derive actionable design guidelines, identifying which methods most effectively enhance real-world reliability and where more research is required.

Addressing the challenge in comparability and verifiably reliable grid operation, our work fills a critical gap by objectively assessing which design choices truly confer systematic generalization for voltage prediction, providing a necessary contribution for the maturity of the field.

**Manuscript Revisions**

Throughout the rebuttal period, we earnestly considered all reviewer comments and have incorporated comprehensive changes, ensuring the paper is now methodologically sound and fully documented, ultimately strengthening our findings. The majority of these additions beyond re-phrasing and re-structuring were made in the Appendix. The following is a summary of manuscript improvements made:

- Clearer definitions and mathematical formulations of the experiment framework, including diagrams of experimental methodology, detailed task formulations, extended dataset descriptions, etc.
- Additional model efficiency metrics such as inference time and model capacity.
- Additional experiments of hybrid inductive bias models created by pairwise combinations of the original inductive biases.
- Ablation studies for cross-architecture and model capacity validation. Here we evaluate a different GNN backbone and capacity-matched models to verify that inductive bias impact remains consistent despite changes to model architecture and number of parameters.
- A secondary analytical baseline, LinDistFlow, against which our physics-informed models continue to boast performance gains.

All core concerns (both methodological and conceptual) have been directly addressed and resolved via concrete revisions and new experimental results. Since we did not receive any reviewer comments, we urge the Area Chair to give careful consideration to our responses, as we believe they fully mitigate the criticisms of the original reviews. We would appreciate any additional support that provides us with the opportunity to present our work at ICLR.

---

### Meta-Review · Area_Chair_CJC3 · 2026-01-07

**Summary:**

Reviewers find the empirical benchmarking of physics-informed strategies for voltage prediction timely and generally well organized, with one reviewer leaning slightly positive, but the overall concerns include the limited conceptual/methodological novelty, questionable or missing motivated classical baselines (notably DC PF and its applicability/interpretation for distribution grids), missing broader comparisons (architectures, capacities, alternative surrogates), and an initially ambiguous framing that collectively weaken the case for acceptance despite the added rebuttal experiments.

**Reviewer Concerns:**

The rebuttal improves clarity on intended contribution (systematic evaluation rather than “discovering” new biases), adds cross-architecture validation, hybrid-bias experiments, efficiency metrics, and a LinDistFlow baseline, but key concerns remain: the paper still remains as a comparative study with limited new technical insight, the baseline story and real-world interpretability, and the overall evidence may not suffice to justify acceptance.

**Reviewer Scores:**

Reviewer M7Jo would likely remain marginally positive given the added math detail and hybrid experiments, while Reviewers M86w and iiq2 would likely stay at reject due to persistent concerns about novelty/positioning and baseline validity; overall, the score distribution would still not reach a clear acceptance consensus after a full discussion.

---

### Decision · Program_Chairs · 2026-01-26

Reject